# Loss-of-function of an *α-SNAP* gene confers resistance to soybean cyst nematode

Mariola Usovsky [1,5], Vinavi A. Gamage [2,5], Clinton G. Meinhardt[1], Nicholas Dietz [1], Marissa Triller[1], Pawan Basnet [1], Jason D. Gillman [3], Kristin D. Bilyeu [3], Qijian Song[4], Bishnu Dhital[1], Alice Nguyen [1], Melissa G. Mitchum [2] ✉ & Andrew M. Scaboo [1] ✉

Plant-parasitic nematodes are one of the most economically impactful pests in agriculture resulting in billions of dollars in realized annual losses worldwide. Soybean cyst nematode (SCN) is the number one biotic constraint on soybean production making it a priority for the discovery, validation and functional characterization of native plant resistance genes and genetic modes of action that can be deployed to improve soybean yield across the globe. Here, we present the discovery and functional characterization of a soybean resistance gene, *GmSNAP02*. We use unique bi-parental populations to fine-map the precise genomic location, and a combination of whole genome resequencing and gene fragment PCR amplifications to identify and confirm causal haplotypes. Lastly, we validate our candidate gene using CRISPR-Cas9 genome editing and observe a gain of resistance in edited plants. This demonstrates that the *GmSNAP02* gene confers a unique mode of resistance to SCN through loss-of-function mutations that implicate *GmSNAP02* as a nematode virulence target. We highlight the immediate impact of utilizing *GmSNAP02* as a genome-editing-amenable target to diversify nematode resistance in commercially available cultivars.

Plant-parasitic nematodes can be devastating to grain and vegetable yield performance across the majority of farmable land and crop species around the world, and the soybean cyst nematode (SCN, *Heterodera glycines* Ichinohe) is consistently the most economically damaging pathogen of soybean (*Glycine max* (L.) Merr.)[1–3]. Estimated yield loss caused by SCN in the United States reached $32 billion between 1996 and 2016, with an average of $1.5 billion annually[4]. Farmers use a multifaceted systems approach to manage parasitic nematodes comprised of growing genetically resistant cultivars in rotation with non-host crops and applying seed treatments before planting to prevent infection and suppress the expansion of population virulence and density[5–8].

Resistance breeding successfully relies on the introgression of major resistance genes (*R*-genes) that indirectly or directly recognize pathogen effectors and confer genetic resistance in crops. Pathogens are under strong negative selection when exposed to these genes resulting in pathotypes that can evolve to overcome resistance. In soybean, native SCN resistance is remarkable in that it employs the disruption of core housekeeping genes largely controlled independently and/or by the interactions of three Mendelian classified genes and their alternate alleles, *Rhg1/rhg1-a* and *rhg1-b*, *Rhg2/rhg2* and *Rhg4/rhg4*[9–12]. The *Rhg1* locus, located on chromosome (Chr.) 18, contains a tandem repeat of three genes present with observable copy number

[1]Division of Plant Science and Technology, University of Missouri, Columbia, MO 65211, USA. [2]Department of Plant Pathology and Institute of Plant Breeding, Genetics and Genomics, University of Georgia, Athens, GA 30602, USA. [3]Plant Genetics Research Unit, United States Department of Agriculture-Agricultural Research Service, University of Missouri, Columbia, MO 65211, USA. [4]Soybean Genomics and Improvement Laboratory, Beltsville Agricultural Research Center, Beltsville, MD 20705, USA. [5]These authors contributed equally: Mariola Usovsky, Vinavi A. Gamage. ✉e-mail: melissa.mitchum@uga.edu; scabooa@missouri.edu

variation (CNV) including an α-soluble NSF (*N*-ethylmaleimide-sensitive factor) attachment protein gene (*α-SNAP*, *GmSNAP18-a*, *GmSNAP18-b*))[9]. The resistance allele *rhg1-b*, derived from plant introduction (PI) 88788, has high CNV, whereas the alternative resistance allele *rhg1-a*, derived from the cultivar Peking, has low CNV[10]. Similarly, at the *Rhg2* locus on Chr. 11, a paralogous *α-SNAP* gene (*GmSNAP11*) has been fine-mapped and shown to confer resistance by an epistatic effect with *rhg1-a*[12–14]. *GmSNAP11* carries a nonsense mutation that causes mis-spliced mRNA that leads to intron retention and translational termination, presumably truncating the protein and resulting in a gain of function in resistance to SCN[13–15]. The third major SCN resistance gene, *Rhg4* (*GmSHMT08*), located on Chr. 8, encodes a cytosolic serine hydroxyl methyltransferase with two amino acid polymorphisms in the ligand binding pocket of the enzyme resulting in a gain of function in resistance to SCN[11].

In total, there are five *α-SNAP* genes located on Chrs. 2, 9, 11, 14, and 18 within the annotated soybean genome (Wm82.a2), yet only the *α-SNAPs* on Chr. 11 and 18 have been reported to function in resistance to SCN[12–14]. The functionality of *α-SNAPs* in eukaryotes is driven by their role in vesicle trafficking through interaction with NSFs for soluble NSF attachment protein receptor (SNARE) complex disassembly at cell membranes[16]. In soybean, it is the local hyperaccumulation of dysfunctional variants of *α-SNAPs* upon nematode infection that is presumed to be disruptive to vesicular trafficking thereby deterring the nematode from establishing an active feeding site resulting in death of this obligate sedentary endoparasite[13,17]. The toxic effect of variant *α-SNAPs* is offset by a stabilizing interaction with a unique NSF during plant growth and development[18], but how the nematode overcomes this resistance remains unknown[19].

The most widely deployed genetic resistance mechanism in soybean, *GmSNAP18-b*, was originally derived from landrace PI 88788 and has been shown to have reduced effectiveness as more virulent nematode populations have arisen as a result of the overutilization of this type of resistance[20,21]. Thus, there is an urgency to identify additional sources of native resistance with unique modes of action for combating these virulent populations that have overcome the aforementioned resistance mechanism[22]. Durable and potentially broadspectrum disease resistance in plants can also be achieved by the loss of functionality of susceptibility genes (*S*-genes)[22]. *S*-genes are classically defined as plant genes present in a susceptible host that are induced and/or targeted by a pathogen for successful infection[23,24]. Many of the *S*-genes identified to promote nematode infection are induced in feeding sites and serve as effector targets[25,26]. The loss-of-function of these *S*-genes results in enhanced nematode resistance[27–29].

In this study, we identified a QTL governing increased resistance to a virulent SCN population on Chr. 2 (QTL02) in soybean and identified the *GmSNAP02* candidate gene within a 218 kb fine-mapped interval. A whole-genome resequencing (WGRS) analysis identified two *GmSNAP02* haplotypes carrying either an insertion or deletion in *GmSNAP02*. CRISPR-Cas9 editing confirmed that *GmSNAP02* confers a unique mode of resistance to SCN through loss-of-function mutations, which also implicates *GmSNAP02* as a potential nematode virulence target.

## Results
### Phenotypic evaluation and linkage mapping
We used three populations: PI 90763 × Peking, Forrest × PI 437654, and SA10-8471 × PI 90763 to map resistance to the widespread virulent SCN HG type 1.2.5.7 population. Resistance to SCN was determined based on the female index (FI), which provides a relative measure of nematode reproduction to susceptible controls. The frequency distribution of FI for all populations is presented in Fig. 1a and Supplementary Fig. 1. The frequencies of individuals for FI did not follow a normal distribution for all three populations which is indicative of qualitative inheritance. In the population PI 90763 × Peking, FIs were determined

to be 0 (resistant; R) and 19.8 (moderately resistant; MR) for PI 90763 and Peking, respectively; and the distribution was skewed towards resistance. In the population Forrest × PI 437654, FIs were determined to be 86 (susceptible; S) and 0.6 (R) for Forrest and PI 437654, respectively; and a bimodal distribution was observed. In the population SA10-8471 × PI 90763, FI were determined to be 88 (S) and 0 (R) for SA10-8471 and PI 90763, respectively; and a bimodal distribution was observed.

A total of 1135, 1541, and 2188 single-nucleotide polymorphisms (SNPs) were utilized to construct linkage maps for populations PI 90763 × Peking, Forrest × PI 437654, and SA10-8471 × PI 90763, respectively (Supplementary Fig. 2). Reduced density of polymorphic SNPs in the first two populations signified strong genomic similarity between the parents. In the population PI 90763 × Peking, we identified a major QTL on Chr. 02 (QTL02) and a minor QTL on Chr. 12 (QTL12) (Fig. 1b and Supplementary Fig. 3). The QTL02 was mapped between Gm02_42,012,522 and Gm02_46,907,259 (Wm82.a2) with a LOD of 8.9, and it accounted for 22.9% of the phenotypic variation. The QTL12 was mapped between Gm12_7791511 and Gm12_9149774 with a LOD of 4.1 and accounted for 9.6% of the phenotypic variation. Beneficial alleles for resistance of both QTL were derived from PI 90763. The same QTL were mapped using RQTL (Supplementary Fig. 4). The ANOVA test showed significance of QTL02 and QTL12 for the peak markers, but interactions between these QTL were not significant (Supplementary Fig. 5). In the population Forrest × PI 437654, we identified a minor QTL on Chr. 2 (QTL02) and a major QTL on Chr. 11 (QTL11) using MapQTL and RQTL (Fig. 1c and Supplementary Figs. 3 and 4). The QTL02 was mapped between the markers Gm02_43,556,059 and Gm02_45,106,877, with the LOD score of 4.4 and $R^2$ of 3.1, and both beneficial alleles of the QTL were derived from PI 437654. The position of QTL02 in this population overlapped with the position of QTL02 in the population PI 90763 × Peking. The QTL11 was mapped between Gm11_32,276,359 and Gm11_33,309,696 with a LOD score of 29.1 and an $R^2$ of 35.9. This QTL overlaps with the position of *GmSNAP11* at the *rhg2* locus[12]. The ANOVA test showed the significance of QTL02 and QTL11 at the peak markers, as well as significant epistatic interactions (Supplementary Fig. 5). Lastly, two major QTL on Chr. 11 (QTL11) and 18 (QTL18) were mapped in the population SA10-8471 × PI 90763 using two mapping methods (Fig. 1d and Supplementary Figs. 3 and 4). The QTL11 was mapped between Gm11_32,586,847 and marker SNAP11-1[30] (Gm11_32,968,127) with a LOD of 22.3 and $R^2$ of 26.9. The QTL18 mapped between Gm18_1427672 and SNAP18-1 with LOD of 15.9 and accounted for 18.2% of the phenotypic variation. Both beneficial alleles of the QTL were derived from PI 90763 and corresponded to the physical position of *GmSNAP11* at the *rhg2* locus, and *GmSNAP18-a* at the *rhg1-a* locus[12]. The ANOVA test showed the significance of the QTL11 and QTL18 peak markers, and significant epistatic interactions between these QTL (Supplementary Fig. 5).

### Fine-mapping of QTL02
In the population PI 90763 × Peking, QTL02 was mapped to a large confidence interval of 4.9 Mb (Fig. 2a and Supplementary Fig. 3). We next investigated recombination events around QTL02 in four populations derived from crosses with PI 90763 as one of the parents (Fig. 2b and Supplementary Fig. 6). In this initial step, we were able to narrow down the QTL interval to a 880 kb (Gm02_44,226,448–45,106,877) region that contained 112 genes based on the reference genome (Wm82.a2). In the next step, we developed a set of 14 KASP assays that span the 880 kb region (Supplementary Fig. 7), and generated $F_{4:5}$ sister lines from the heterozygous lines derived from PI 90763 × Peking that carried recombination events within QTL02. This allowed us to fine-map the region to 218 kb between markers Gm02-09 (Gm02_44,617,603) and Gm02-14 (Gm02_44,835,549) (Fig. 2b and Supplementary Fig. 8). This region contained 34 candidate genes including the *GmSNAP02* gene

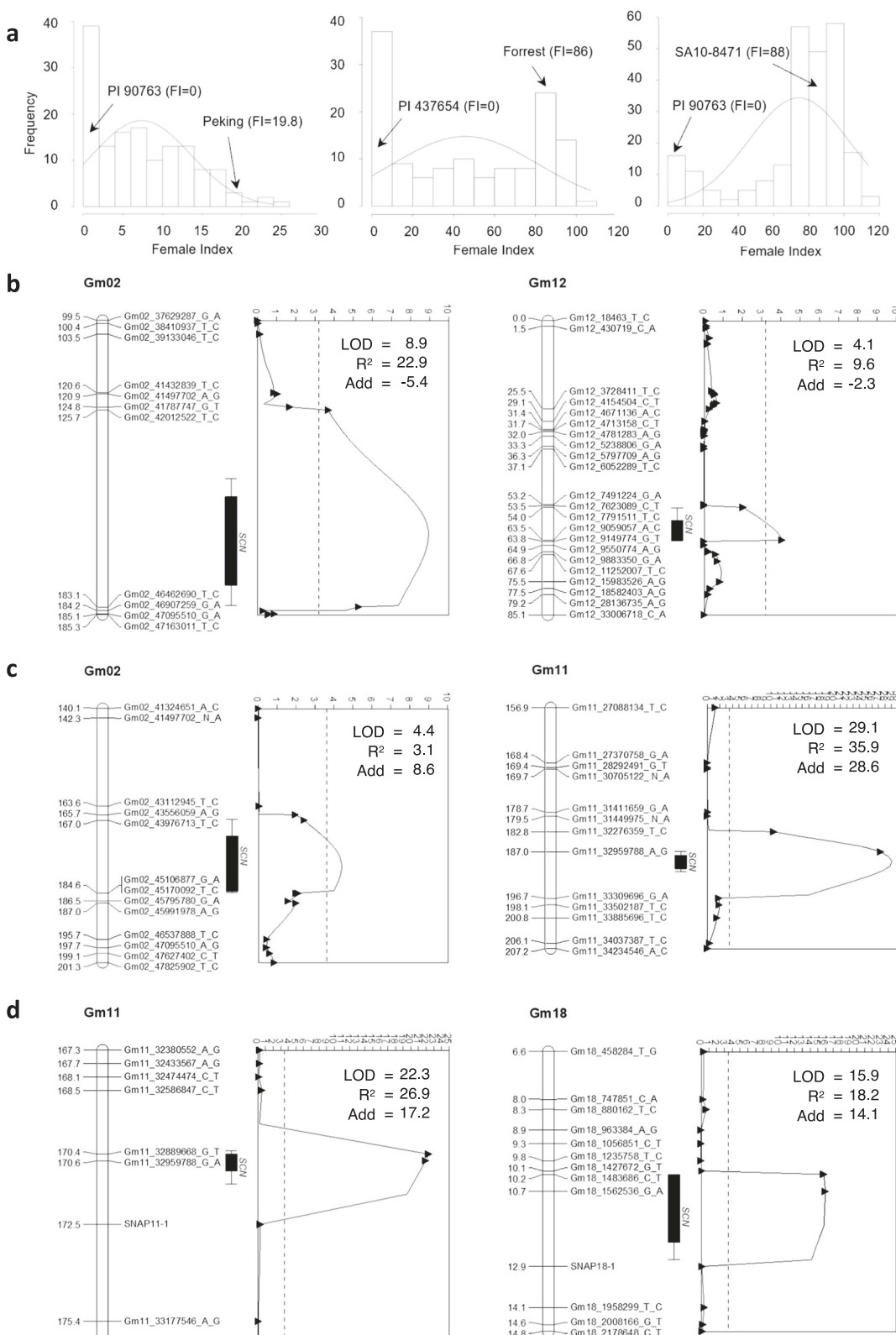

**Fig. 1 | Mapping of quantitative trait loci (QTL) controlling soybean cyst nematode (SCN) HG type 1.2.5.7 resistance. a** Frequency distribution of female indices (FI) in 144 $F_{3:4}$ lines from PI 90763 × Peking, 131 $F_{3:4}$ lines from Forrest × PI 437654, and 244 $F_{3:4}$ lines of SA10-8471 × PI 90763. **b** QTL02 and QTL12 detected in PI 90763 × Peking population. **c** QTL02 and QTL11 detected in Forrest × PI 437654 population. **d** QTL11 and QTL18 detected in SA10-8471 × PI 90763 population. Scales on the top of the graph represent the value of the logarithm of the odds (LOD). The black dotted line indicates the threshold of significance (LOD = 3.2, 3.6, and 3.5) for each population, respectively. Add = additive effect.

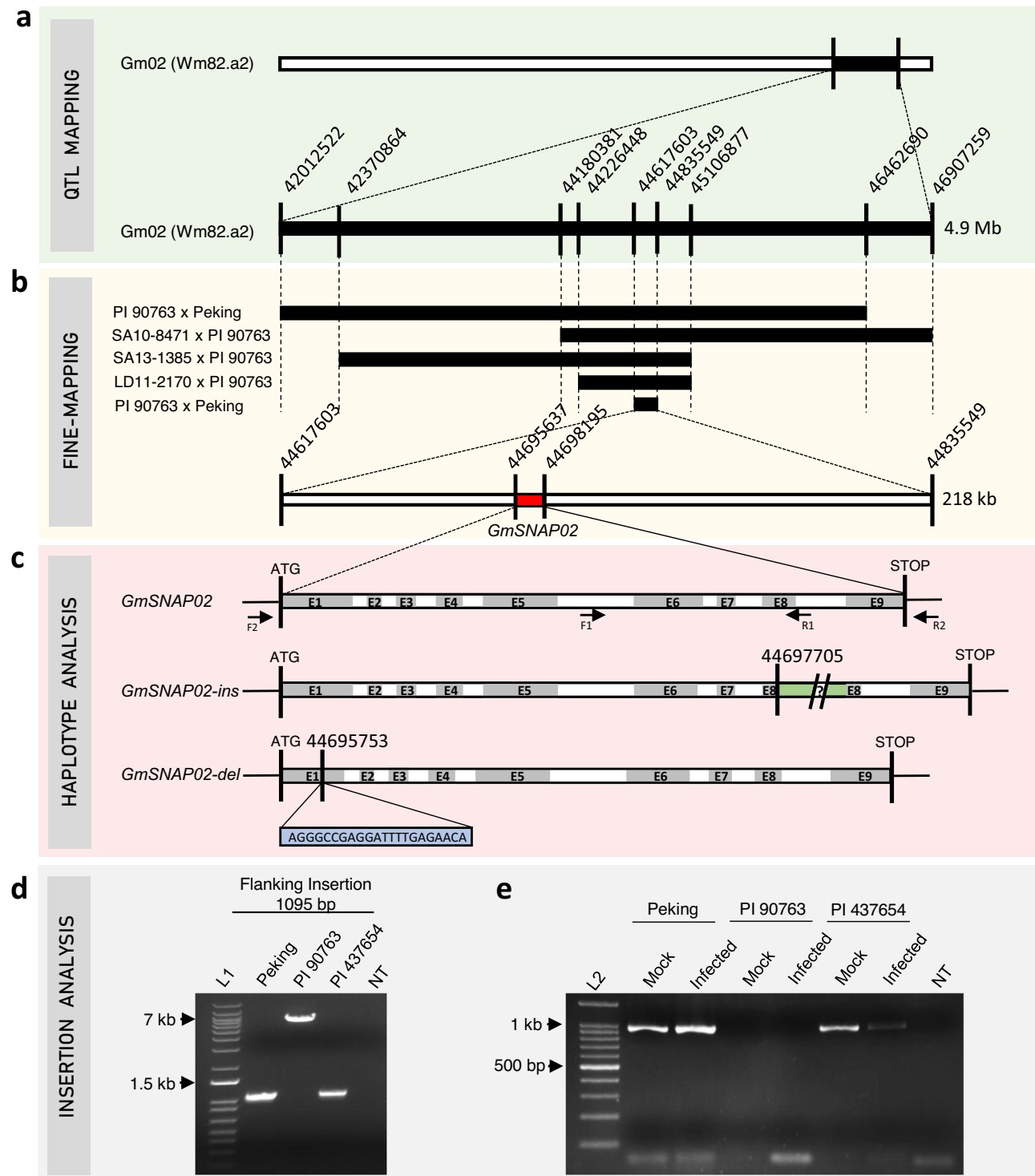

**Fig. 2 | *GmSNAP02* positional cloning. a** The QTL02 confidence interval Gm02: 42,012,522–46,907,259 (Wm82.a2) was identified from a cross between PI 90763 × Peking. **b** An initial fine-mapped region of ~880 kb containing 112 genes was determined using recombinant inbred F$_{3:4}$ lines derived from four populations. F$_{4:5}$ lines derived from the cross between PI 90763 x Peking were used to further narrow the region to ~218 kb containing 34 genes. Within this region *GmSNAP02* (*Glyma.02G260400*) became a candidate gene (red). **c** Haplotypes identified at *GmSNAP02* included the susceptible *GmSNAP02* haplotype of the Williams 82 reference genome, *GmSNAP02-ins*, a resistant haplotype caused by an ~6 kb insertion (green) in exon 8 in PI 90763, and *GmSNAP02-del* haplotype caused by a 22 nt deletion (blue) in exon 1 resulting in a frameshift mutation leading to a premature stop. *GmSNAP02* gene-specific primers designed to flank the insertion in exon 8 (**d**, F1/R2) amplified products of the predicted size in Peking and PI 437654, but a larger product in PI 90763, confirming the presence of an insertion. This experiment was repeated four times with similar results. **e** *GmSNAP02* gene-specific primers designed immediately upstream of the start and downstream of the stop codons (**c**, F2/R2) amplified a full-length *GmSNAP02* cDNA sequence from mock-inoculated and SCN-infected (3 days post inoculation) roots of Peking and PI 437654, but not from PI 90763. Sequencing of the product amplified from PI 437654 confirmed the presence of a 22 nt deletion in exon 1 of *GmSNAP02*. This experiment was repeated twice with similar results. L1 = 1 kb Plus ladder (Invitrogen), L2 = 100 bp ladder (NEB), NT = no template control.

(*Glyma.02G260400*) that encodes an *α-SNAP* protein. The *GmSNAP02* gene is a paralog of *GmSNAP18* at *Rhg1* and *GmSNAP11* at *Rhg2*, both contributing to SCN resistance[12–14,31], thus, we considered the *GmSNAP02* to be a major candidate gene at QTL02.

## Whole-genome resequencing of the *GmSNAP02* gene

We utilized the Soybean Allele Catalog Tool (SAC) to obtain the series of gene-modifying haplotypes present in our gene of interest, based on the set of whole-genome resequencing (WGRS)-derived variants present in a diversity panel of >1000 wild and domesticated soybean accessions[32]. Although the SAC showed 11 potential haplotypes in the *GmSNAP02* gene, both PI 90763 and Peking had the reference call of Williams 82 at each position, and thus exhibited no obvious polymorphic sites (Supplementary Fig. 9). However, two of the eleven variant positions (Chr02_44,697,698 and Chr02_44,697,700) originally lacked genotype calls in PI 90763 and were imputed as reference calls, whereas Peking had confident reference calls at each of the 11 variant positions prior to imputation. To investigate these lines further for other potential sources of variation, we explored the WGRS-derived variant call data across the entire *GmSNAP02* gene, as far upstream as 20 kb from the transcription start site, but still found no polymorphisms between PI 90763 and Peking. Moreover, raw reads of WGRS were used to analyze copy number variation (CNV) of *GmSNAP02* across over 1000 soybean accessions, and variations were not observed for *GmSNAP02*, whereas CNV was confirmed for *GmSNAP18* (Supplementary Fig. 10).

Next, we mapped the raw sequencing reads back to the Williams 82 reference genome and visually inspected the reads aligned to the *GmSNAP02* gene. This analysis confirmed that there were no SNPs or small insertions or deletions (Indels) present in the genic region between PI 90763 and Peking. However, we observed a pattern of reads with poor alignment in the eighth exon of the *GmSNAP02* gene in PI 90763 (Supplementary Fig. 11a). This pattern was absent from the alignment data for Peking, PI 437654, and Williams 82. We posited that this pattern was indicative of an insertion that is too large to be accurately characterized by the short-read sequencing data utilized in our analysis (Supplementary Fig. 11b).

## Haplotypes *GmSNAP02-ins* and *GmSNAP02-del*

To further confirm the presence of an insertion in *GmSNAP02* in PI 90763, gene-specific primer sequences F1/R1 were designed flanking the predicted insertion in the eight exon of PI 90763 (Fig. 2c and Supplementary Fig. 12) and were used to amplify the corresponding *GmSNAP02* sequence. A product of the expected size (1095 bp) was amplified from genomic DNA of Peking and PI 437654, but a larger product was amplified from PI 90763, confirming an ~6 kb insertion in *GmSNAP02* (Fig. 2d). In addition, gene-specific primers F2/R2 designed immediately upstream of the start and downstream of the stop codons (Fig. 2c and Supplementary Fig. 12) were used to amplify full-length *GmSNAP02* transcripts from cDNA generated from total RNA isolated from both mock and SCN-infected root tissues at 3 days post inoculation (dpi). A product of the expected full-length *GmSNAP02* transcript size (949 bp) was amplified from Peking and PI 437654, but not from PI 90763 (Fig. 2e). Our inability to amplify *GmSNAP02* transcripts in PI 90763 may be attributed to its large size (~ 7 kb) and/or nonsense-mediated mRNA decay. These results further confirmed the presence of an insertion in PI 90763. A haplotype *GmSNAP02-ins* was designated for an insertion in exon 8 in PI 90763. The graphical representation of this haplotype is depicted in Fig. 2c.

Aligning the raw reads of WGRS did not detect a similar insertion in exon 8 in *GmSNAP02* of PI 437654 (Supplementary Fig. 11a). However, the SAC revealed another haplotype containing a deletion of 22 nucleotides (5'-AGGGCCGAGGATTTTGAGAACA-3') at the physical position Gm02_44,695,753 (Wm82.a2) in the first exon of the gene. This deletion results in a frameshift beginning at the eighth amino acid.

A haplotype *GmSNAP02-del* was designated for a deletion in exon 1 in PI 437654. The graphical representation of this haplotype is depicted in Fig. 2c. *GmSNAP02* gene-specific primers were designed to amplify a full-length *GmSNAP02* cDNA sequence from mock-inoculated and SCN-infected (3 dpi) roots. The products were amplified and sequenced from Peking and PI 437654 using primer set F2/R2 and the presence of a 22-nucleotide deletion in exon 1 in PI 437654 of *GmSNAP02* causing a frameshift mutation leading to a premature stop codon was confirmed (Fig. 2e and Supplementary Fig. 13).

## *GmSNAP02* expression in response to SCN infection

To monitor the early stages of SCN infection, soybean roots were stained with acid fuchsin at 3- and 5 dpi to visualize nematode life stages within roots. By 3 dpi, infective second-stage juveniles (J2) were observed inside the roots of all lines (Fig. 3a, top panel). At 5 dpi, a subset of sedentary, swollen J2 and J3 life stages representing feeding nematodes that had advanced in their development were observed in Peking, but not PI 90763 and PI 437654 (Fig. 3a, bottom panel). These nematodes reflect the portion of the HG type 1.2.5.7 population that will ultimately give rise to the ~20% female index on Peking and also differentiates the virulent populations HG type 2.5.7 and HG type 1.2.5.7 (Fig. 1a). Therefore, the expression of *GmSNAP02* in response to nematode infection was evaluated at 3 dpi by real-time qRT-PCR analysis of uninfected (mock) and infected roots of Peking, PI 90763, PI 437654 and two residual heterozygote-derived lines near-isogenic for *GmSNAP02*, 19AS-84-5-81-4 (81-4) and 19AS-84-5-81-8 (81-8), which are moderately resistant (FI = 16) and highly resistant (FI = 0) to HG type 1.2.5.7, respectively (Supplementary Fig. 14). Peking and 81-4 showed consistent and significant upregulation of *GmSNAP02* across three and two biological replicates, respectively, whereas PI 90763, 81-8, and PI 437654, either showed significant downregulation or no differential regulation of *GmSNAP02* in response to nematode infection (Fig. 3b). Taken together, these results suggested that an upregulation of *GmSNAP02* in response to SCN infection in Peking promotes susceptibility, whereas the deleterious effects of *GmSNAP02-ins and GmSNAP02-del* on the expression of *GmSNAP02* in PI 90763 and PI 437654 in response to SCN infection enables plant resistance.

## *GmSNAP02* loss-of-function confers SCN HG type 1.2.5.7 resistance

To directly test whether a loss-of-function of *GmSNAP02* results in SCN resistance, we used a CRISPR/Cas9 dual guide system for targeted knockout of *GmSNAP02* in transgenic hairy roots of Peking composite plants[33,34]. Two dual CRISPR/Cas9 constructs were designed to target the 5' and 3' regions of the *GmSNAP02* gene for deletion. CRISPR/Cas9-*GmSNAP02*-T4 + T3 was designed to make double-strand breaks in exon 1 and exon 5 that were 871 bp apart and CRISPR/Cas9-*GmSNAP02*-T5 + T7 was designed to target two sites in exon 9 and the 3'-UTR region that were 37 bp apart (Fig. 4a). Composite soybean plants of Peking, and PI 90763 for comparison, were generated by co-cultivation with *Rhizobium rhizogenes* K599 carrying the respective CRISPR/Cas9-*GmSNAP02*-gRNAa+gRNAb construct and an empty vector (EV) control. Plants with transgenic roots selected by positive green fluorescing protein (GFP) were used in SCN bioassays (Fig. 4b and Supplementary Figs. 15–18) and twenty-eight days postinoculation cysts extracted from roots were counted under a stereoscope. Peking roots transformed with CRISPR/Cas9-*GmSNAP02*-T4 + T3 and CRISPR/Cas9-*GmSNAP02*-T5 + T7 showed a significant reduction in cyst numbers compared to plants transformed with EV in two independent biological replicates (Fig. 4c and Supplementary Figs. 15 and 17). PI 90763 did not show a significant difference in cyst counts between plants transformed with EV and CRISPR/Cas9-*GmSNAP02*-T4 + T3 or -T5 + T7 (Fig. 4c and Supplementary Figs. 16 and 18). Transgenic roots from a subset of plants from each genotype/construct were collected to confirm CRISPR/Cas9-induced edits in *GmSNAP02*. *GmSNAP02*

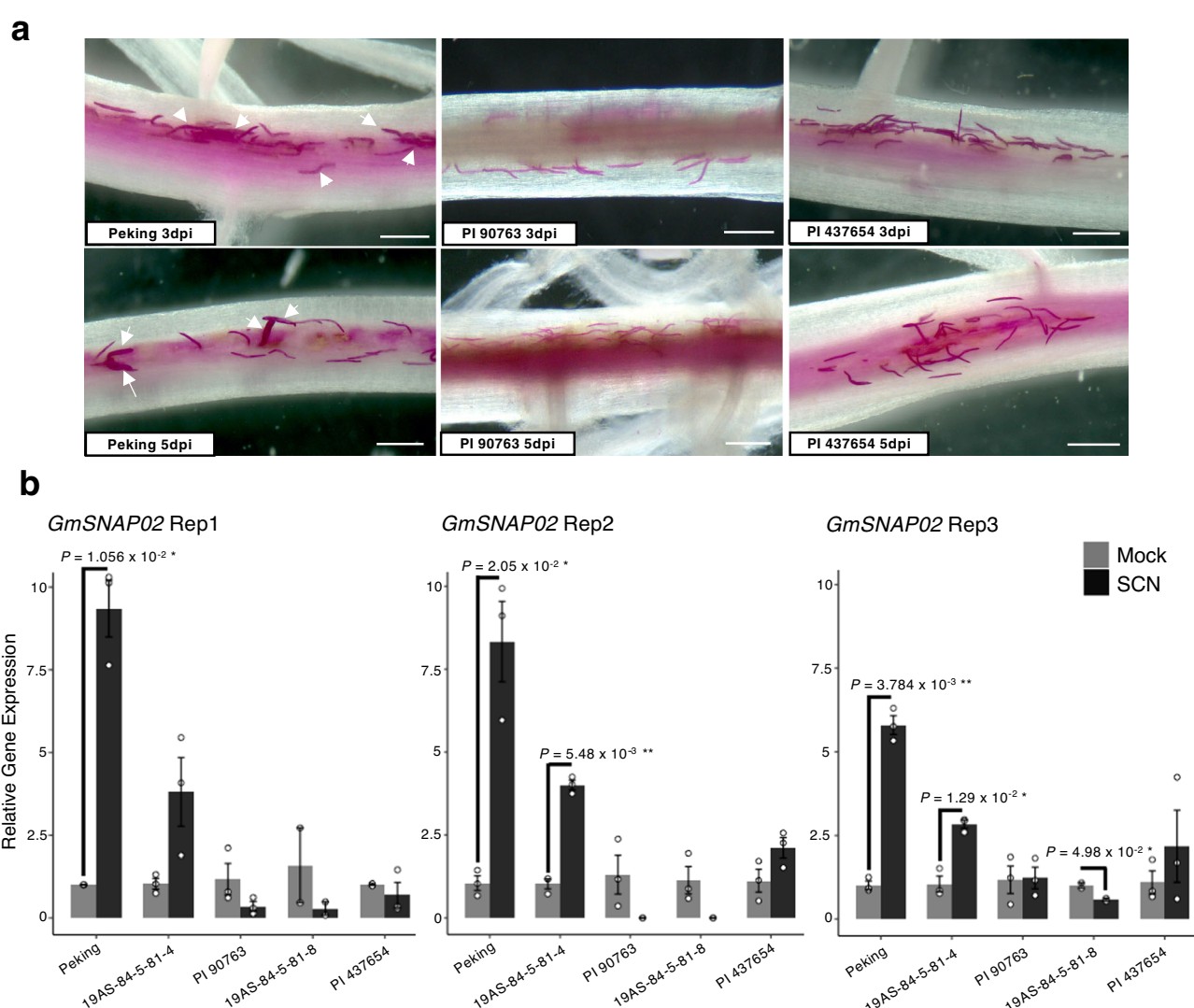

**Fig. 3 | GmSNAP02 gene expression in response to SCN infection. a** Nematode development on Peking, PI 90763, and PI 437654 soybean lines. Roots of 3-day-old seedlings were inoculated with infective second-stage juveniles (J2) and stained with acid fuchsin at 3 and 5 days post inoculation (dpi). Representative images of 3–5 independent roots/genotype are shown. White arrowheads denote swollen parasitic juvenile nematodes in Peking indicative of successful feeding site establishment and development. Scale bar = 500 μm. **b** Relative gene expression analysis of *GmSNAP02* in mock-inoculated and SCN-infected roots using qRT-PCR. Two residual heterozygote-derived lines near-isogenic for *GmSNAP02*, 19AS-84-5-81-4 and 19AS-84-5-81-8, were also included. Statistical analysis was performed by a two-tailed Student's *t* test. Data are means ± s.e.m. of three technical replicates for each biological replicate. Data from three independent biological replicates showed a significant (*$P < 0.05$, **$P < 0.01$) increase in *GmSNAP02* gene in expression in susceptible Peking and 81-4, but not PI 90763, 81-8, and PI 437654 upon SCN HG type 1.2.5.7 (Race 2) infection.

sequence-specific primer pairs F3-R3 or F4-R4 (Fig. 4a), flanking the target sites, amplified the expected 1160-bp and 466-bp long fragments from plants transformed with EV, respectively (Fig. 4d). However, not all plants transformed with the *GmSNAP02* CRISPR/Cas9 constructs amplified the expected 891 bp and 37 bp fragments to indicate full deletions (Fig. 4a, d). Thus, deletion events were further confirmed in selected plants by amplicon sequencing and *GmSNAP02*-specific edits were confirmed in both Peking and PI 90763 (Supplementary Figs. 19 and 20). The absence of unintended edits in *GmSNAP14*, a closely related paralogous gene, was confirmed through site-specific amplification and sequencing of the predicted off-target sites in selected plants using primers F5-R5 and F6-R6 (Supplementary Figs. 21 and 22). Figure 4e shows *GmSNAP02* edits in five plants/construct and the corresponding cyst counts. All *GmSNAP02*-edited roots of Peking exhibited a reduced number of cysts while un-edited Peking plants #6 and #3 transformed with CRISPR/Cas9-*GmSNAP02*-T4 + T3 and T5 + T7, respectively, had higher cyst counts similar to the Peking

EV plants (Fig. 4c, e). No abnormal root growth phenotypes were observed due to the knockout of *GmSNAP02* in either genotype (Fig. 4b and Supplementary Figs. 15–18). Taken together, these results unequivocally demonstrate that a loss-of-function of *GmSNAP02* enhances the resistance of Peking to the virulent SCN HG type 1.2.5.7.

**Response of specific allele combinations to SCN infection**

To investigate the effectiveness of *GmSNAP02* on resistance to various SCN populations (Supplementary Fig. 23), we selected lines from four mapping populations derived from crosses with PI 90763 as one of the parents. These lines contained various combinations of homozygous alleles for all currently known SCN resistance alleles within PI 90763 and/or PI 88788: *GmSNAP18-a*, *GmSNAP18-b*, *GmSNAP11*, *GmSHMT08*, and *GmSNAP02-ins*. Responses of lines with allele combinations containing *GmSNAP18-a* to infection with four SCN populations are depicted in Fig. 5. Supplementary Fig. 24 depicts all allele combinations found in each mapping population. We identified a significant

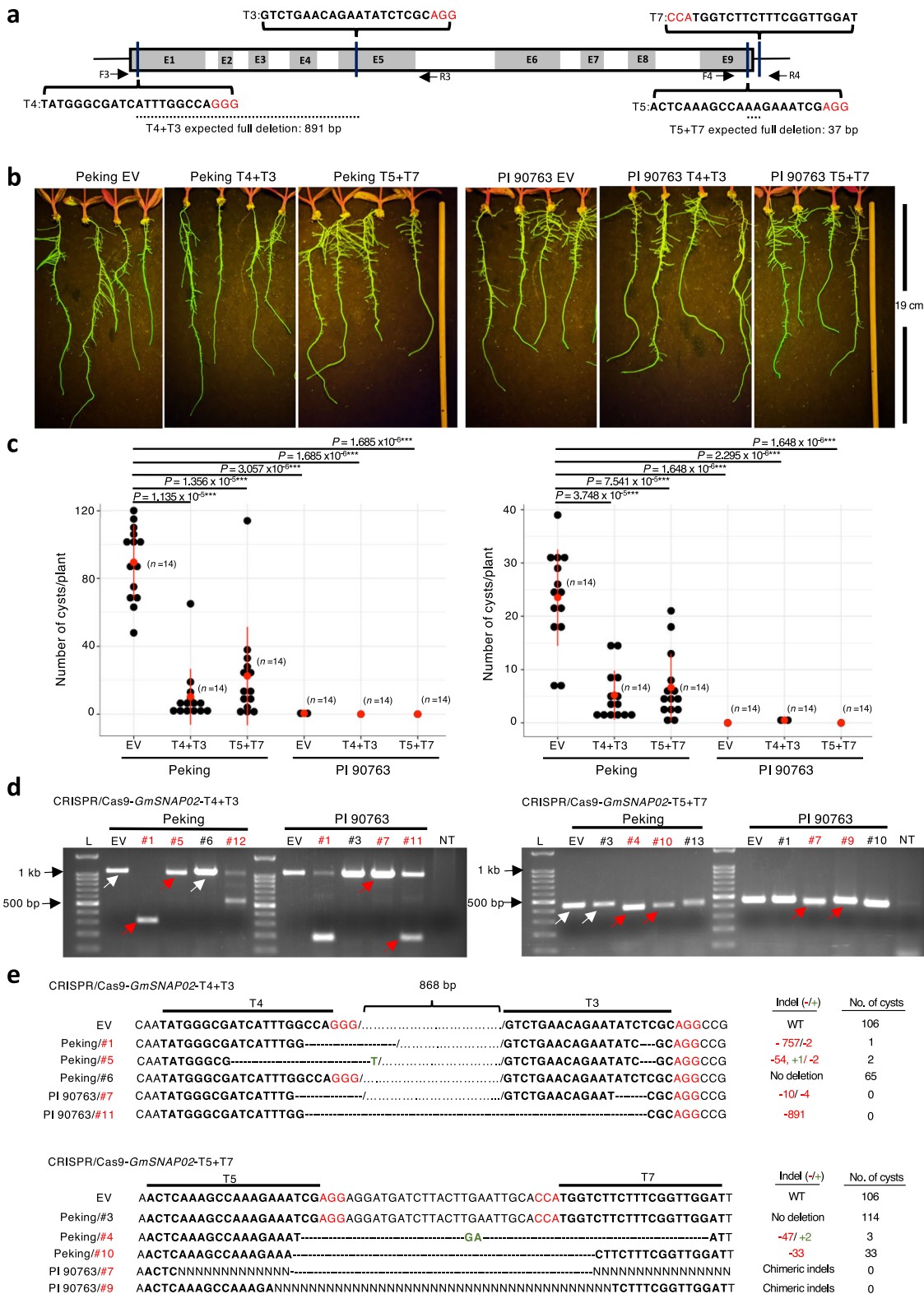

impact of the *GmSNAP18-a* allele on resistance only when combined with other alleles, whereas gene pyramiding with *GmSNAP18-b* was ineffective in conferring resistance (FI < 30) for all three virulent populations. In general, the results indicate that the *GmSNAP02-ins* allele function is dependent on the gene stack *GmSNAP18-a + GmSNAP11*, additionally validating QTL mapping results, for virulent populations SCN HG type 2.5.7 and HG type 1.2.5.7. Adding *GmSNAP02-ins* to the gene stack *GmSNAP18-a + GmSNAP11* caused a shift from moderate resistance to high resistance with FI equal to or near zero (Fig. 5). The addition of *GmSHMT08* to the gene stack *GmSNAP18-a + GmSNAP11 + GmSNAP02-ins* was not necessary to maintain low FI for HG type 1.2.5.7; however, we observed that *GmSNAP18-a + GmSNAP11 + GmSHMT08* plays a role in high resistance to HG types 2.5.7 as previously reported[12]. The gene stack *GmSNAP18-a + GmSNAP11 + GmSNAP02-ins* was the only combination that caused the FI to drop to zero with infection of the HG 1.2.5.7 population. We

**Fig. 4 | Functional validation of *GmSNAP02* in resistance to SCN using CRISPR/ Cas9. a** Diagram showing the positions of the guide RNA (gRNA) sequences designed to edit the *GmSNAP02* gene. PAM sequences are in red. gRNA sequences are bolded. **b** Composite soybean plants with transgenic GFP-positive hairy roots were selected under fluorescent light. Representative images are shown. No gross phenotypic differences were observed in *GmSNAP02*-edited roots of either Peking or PI 90763 (*n* = 14 plants/construct examined in two independent experiments). Pictures taken just before transplanting and nematode inoculation. **c** Cyst counts on transgenic roots of Peking and PI 90763 composite plants transformed with K599 carrying empty vector (EV; control), CRISPR/Cas9-*GmSNAP02*-T4 + T3, and CRISPR/Cas9-*GmSNAP02*-T5 + T7 constructs, respectively. Data are shown for two biological replicates for each construct and genotype. Means ± s.d. are denoted with a red dot and line (*n* = 14). ***P* < 0.001 and ***P* < 0.01, Wilcoxon rank-sum

statistical test. **d** Amplified fragments from genomic DNA extracted from roots transformed with CRISPR/Cas9-*GmSNAP02*-T4 + T3, and CRISPR/Cas9-*GmSNAP02*-T5 + T7, respectively. Fragments flanking T4 and T3 gRNA cleavage sites were amplified using F5 and R5 primers (shown in (**a**)). Fragments flanking T5 and T7 gRNA cleavage sites were amplified using F6 and R6 primers (shown in (**a**)). A subset of roots from both genotypes for each construct was selected for genotyping. Fragments indicated by the arrows were gel extracted, subcloned, and sequenced to confirm deletions. White arrows no deletion, Red arrows/#'s deletion confirmed, L 100 bp ladder (NEB), NT no template control. **e** Sequences of selected fragments from **d**. The number of nucleotides deleted (red font) and/or inserted (green font) and the corresponding cyst counts for each plant are indicated in the columns on the right.

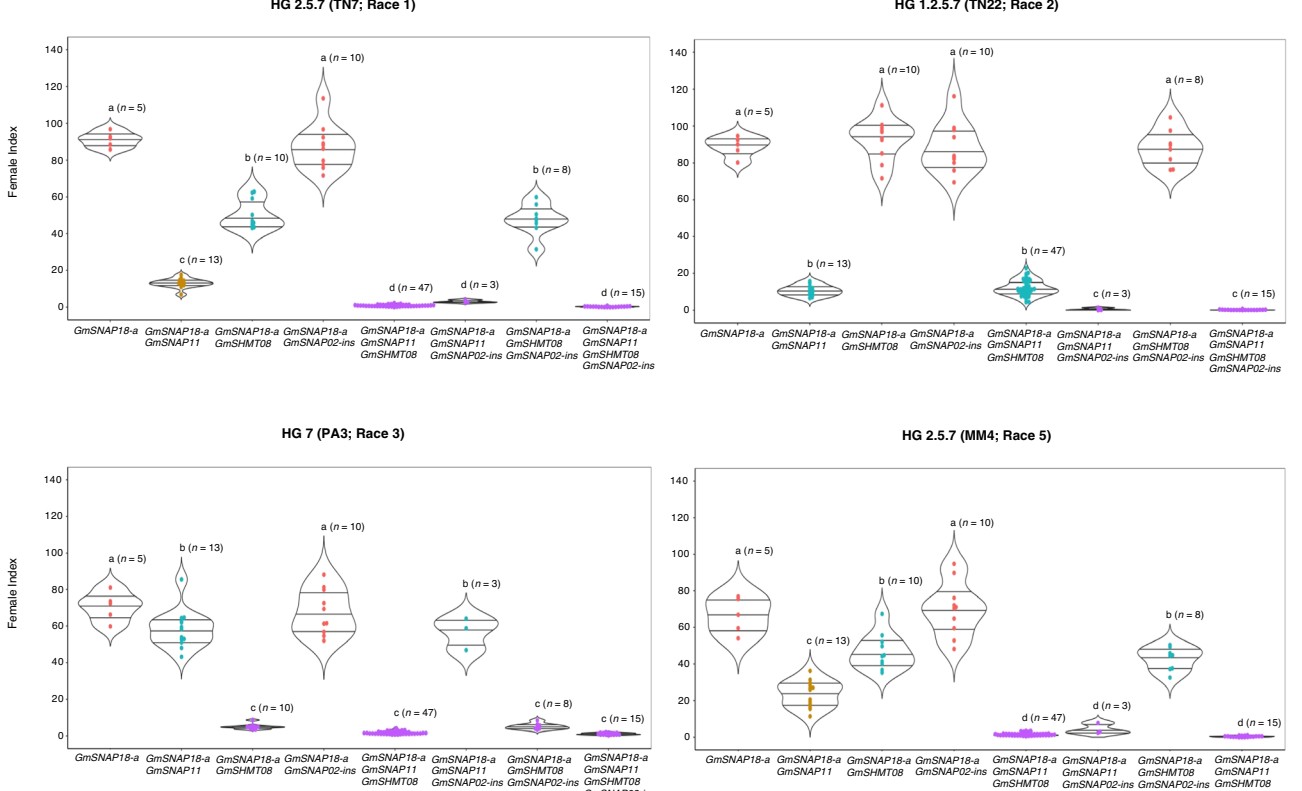

**Fig. 5 | Female index values of pooled F$_{3:5}$ lines with different allelic combinations for four SCN populations.** Lines were pooled from four populations: PI 90763 × Peking, SA10-8471 × PI 90763, SA13-1385 × PI 90763, and LD11-2170 × PI 90763. The numbers in parenthesis above each violin plot correspond to the number of independent lines phenotyped in the pooled genotypic class (*n*). Female indices derived from lines infected from each SCN population were analyzed by

one-way analysis of variance (ANOVA), and Tukey's HSD test was used for multiple comparisons. The ANOVA *P* values were significant for all four individual SCN population tests (*P* ≤ 0.05). The letters above each violin plot correspond to significance/non significance between allelic combination groups based on the Tukey's HSD test (*P* ≤ 0.05). Exact *P* values are provided in the source data file for both ANOVA and Tukey's HSD tests.

observed no significant effect of *GmSNAP02-ins* on the HG 7 population, as both gene stacks *GmSNAP18-a* + *GmSNAP11* and *GmSNAP18-a* + *GmSNAP11* + *GmSNAP02-ins* resulted in susceptibility. The gene stack *GmSNAP18-a* + *GmSHMT08* or a single allele *GmSNAP18-b* were the most efficient in controlling the HG type 7 population (Supplementary Fig. 24).

### High-throughput detection of *GmSNAP02* haplotypes

We utilized the sequences of the mismatched short-read pattern caused by the insertion in exon 8 in PI 90763 to develop a high-throughput diagnostic assay for genotyping and ultimately for marker-assisted breeding (Supplementary Fig. 25a). Two TaqMan assays, MU-SNAP02$^{INS}$-WT and MU-SNAP02$^{INS}$-MUT, were designed for the detection of the *GmSNAP02-ins* haplotype in PI 90763

(Supplementary Fig. 25b, c). The assay MU-SNAP02$^{INS}$-WT consists of a pair of unlabeled PCR primers that align to the sequences flanking the insertion and amplifies an amplicon of 121 bp from the wild-type (WT) *GmSNAP02* in genotypes such as Williams 82 and Peking. The second assay MU-SNAP02$^{INS}$-MUT consists of a forward primer that aligns to the 3' end of the insertion sequence, and a reverse primer that aligns to the sequence flanking the insertion. It amplifies a fragment of 133 bp in mutant (MUT) genotype PI 90763 thereby detecting the allele with the insertion in exon 8. Both assays contain two TaqMan probes of the same sequence labeled with a fluorescent dye FAM and VIC on the 5' end and a minor groove binder (MGB) and non-fluorescent quencher (NFQ) on the 3' end. Having the same probe labeled with different dyes enabled the detection of heterozygotes (HET) and requires running both assays. MU-SNAP02$^{INS}$-WT

amplifies WT and HET genotypes, whereas MU-SNAP02[INS]-MUT amplifies MUT and HET genotypes.

Two TaqMan assays, MU-SNAP02[DEL]-1 and MU-SNAP02[DEL]-2 were developed to differentiate three different variants at the location of the frameshift (Supplementary Fig. 26). The MU-SNAP02[DEL]-1 separates the alternative (to the reference genome Williams 82) allele from the deletion allele (*GmSNAP02-del*) and the MU-SNAP02[DEL]-2 assay determines the reference allele from the deletion allele (*GmSNAP02-del*). Assays MU-SNAP02[INS]-WT, MU-SNAP02[INS]-MUT, MU-SNAP02[DEL]-1, and MU-SNAP02[DEL]-2, along other known SCN assays, were used for genotyping a set of SCN-resistant soybean accessions including indicator lines used in race and HG type protocols (Supplementary Fig. 27).

## Discussion

This study discovered, confirmed, and validated the role of the *GmSNAP02* gene, a paralog of the *GmSNAP18* and *GmSNAP11* genes, in resistance to SCN. This discovery was achieved by designing unique cross-combinations using SCN-resistant parents with the same alleles for known major genes. The detection of QTL02 was only possible by eliminating the large variance in FI caused by the genetic effect of both *GmSNAP18* and *GmSNAP11*. The absence of QTLs on Chrs. 8, 11, and 18 in the population PI 90763 × Peking confirms that PI 90763 and Peking carry the same alleles of *GmSNAP18, GmSNAP11*, and *GmSHMT08*. Thus, we were able to attribute the remaining FI variance caused by virulent SCN to QTL02. To our knowledge, no QTL has been detected in this genomic region, and this is the first study to demonstrate the imperative role of *GmSNAP02* in SCN resistance. In fact, prior studies have concluded that *GmSNAP02* does not contribute to SCN resistance, which was likely a result of the large genetic effects caused by *GmSNAP18* and *GmSNAP11* in conventional resistant-by-susceptible bi-parental mapping populations coupled with the use of an SCN HG type 0 population[14]. Our results show that all three paralogous genes, *GmSNAP18*, *GmSNAP11*, and *GmSNAP02*, contribute to resistance with different genetic modes of action and varying phenotypic outcomes depending on the allele combination and SCN population.

As more complex gene interactions are discovered, it highlights that the full scope of broad and durable SCN resistance remains to be elucidated. It has become increasingly clear that not all SCN resistance genes work well together. The *GmSNAP18-b* haplotype, derived from PI 88788, is effective against SCN HG type 0 and HG type 7 populations, and has been bred into thousands of soybean cultivars and is currently deployed in ~95% of the entire North American market[21]. Consequently, *GmSNAP18-b* has been used by soybean farmers for decades on tens of millions of acres as the predominant source of SCN resistance. Though a combination of *GmSNAP18-a* and *GmSHMT8* is equally effective against SCN HG type 0 and HG type 7 populations and has been used to develop SCN-resistant cultivars, this resistance only represents 5% of the market[35], making the current management recommendation to include a rotation of these two types of resistance impractical for most growers and regions of high productivity. Moreover, recent surveys show a disturbing and widespread increase in SCN populations virulent on these two types of resistance[20,35–37]. Unfortunately, efforts to improve upon *GmSNAP18-b* resistance using conventional breeding approaches have been unsuccessful because it does not confer enhanced resistance when pyramid with other known resistance alleles[12]. On the other hand, a recent study demonstrated that *GmSNAP18-a* must be combined with *GmSNAP11* to impart resistance to virulent SCN populations that have overcome the *GmSNAP18-b* alleles' resistance mechanism[12]. Our discovery of *GmSNAP02* and an analysis of the response of specific allele combinations to SCN showed evidence that *GmSNAP18-a* and *GmSNAP11* are necessary to observe the phenotypic effect of *GmSNAP02*. Thus, the amassing evidence for the importance of pyramiding specific allele combinations with the *GmSNAP18-a* haplotype also suggests that the quadruple allele stack

*GmSNAP18-a* + *GmSNAP11* + *GmSNAP02* + *GmSHMT08* will provide effective and broad-spectrum resistance to SCN populations.

Here, we provided multiple lines of evidence that a loss-of-function of *GmSNAP02* confers resistance to nematodes that can overcome resistance mediated by *GmSNAP18-a* + *GmSNAP11*. This is in stark contrast to our current understanding of the mechanism of *GmSNAP18-a/-b* type resistance. *GmSNAP18s* code for atypical SNAPs defined by polymorphisms in C-terminal residues at a conserved functional site[9]. The expression of *GmSNAP18* increases upon nematode infection leading to hyperaccumulation of atypical SNAPs tipping the balance of endogenous α-SNAP-NSF interactions and triggering cytotoxicity at nematode feeding sites due to the disruption to vesicular trafficking[17]. In SCN-resistant genotypes, atypical *GmSNAP18* protein exhibits stronger binding to a corresponding atypical compatible NSF protein variant encoded by the gene *GmNSF_RAN07*. In this way, the SNARE recycling machinery abrogates *GmSNAP18* cytotoxicity, thus ensuring plant fitness and viability[18]. As an obligate, sedentary endoparasite, SCN modifies host cells into a permanent feeding site in host roots. For this, SCN delivers a cocktail of effector proteins through a stylet into a selected host cell near the vascular cylinder. These effector proteins function to suppress host immunity and co-opt a wide array of host cellular processes to create a multinucleate and highly metabolically active cell type that serves as a nutrient sink[38]. Considering the critical roles of the vesicular trafficking pathway in a range of plant developmental and stress-responsive pathways, it stands to be a prime target for nematode effectors to exploit for successful parasitism[39]. Thus, plants have likely evolved to mount a local counterattack by evading recognition of these effectors to block feeding site formation. In line with this, our findings point to *GmSNAP02* as a possible virulence target of nematode effectors, as only a loss-of-function mutation will enable resistance for the plant.

Although host targets are exploited by pathogens to promote disease, their mutations can lead to durable, recessively inherited, and potentially broad-spectrum resistance in plants[22]. To date, these so-called susceptibility genes (*S*-genes) have been primarily identified through forward genetic studies. Here, we also used forward genetics, followed by gene editing for targeted knockout of the *GmSNAP02* gene in the cultivar Peking, confirming this gene as a new and vitally important *S*-gene in soybean. The genetic diversity of *S*-genes is currently understudied; however, several natural mutant alleles in multiple crop species have been identified[40–42]. Here we identified two different native mutant *GmSNAP02* gene haplotypes, including an ~6 kb insertion, and a 22 bp deletion leading to a frameshift mutation causing a premature stop codon. Although the exact nature of the insertion awaits further analysis, we hypothesize a transposon insertion is a likely possibility. The *GmSNAP18-a* also carries a retro-transposon in the first intron, while the *GmSNAP18-b* and *GmSNAP18-c* (WT) do not[43]. Moreover, the haplotype *GmSNAP02-ins* is also present in PI 507471, and the haplotype *GmSNAP02-del* is present in PI 89772, the cultivar Hartwig (PI 543795), the cultivar S05-11482, PI 417091, PI 404166, and PI 567336B[24].

In summary, *GmSNAP02* can be applied in an improved breeding and biotechnology strategy to significantly enhance soybean plant resistance to SCN. The introgression of native mutant haplotypes of *GmSNAP02-ins* from PI 90763 and *GmSNAP02-del* from PI 437654 into elite soybean cultivars that carry the *GmSNAP18-a* + *GmSNAP11* gene stack is straightforward and easily attainable. Moreover, a transgene-free system of precise genome editing of *GmSNAP02* could also be implemented for improving SCN resistance in modern cultivars, and to our knowledge this is the first viable target for such an application in soybean. The CRISPR/Cas9 gene editing system has already been used in the study of multiple *S*-genes[44–48]. The enhancement of soybean resistance using *GmSNAP02* in a gene pyramiding scheme provides a pragmatic and straightforward solution to diversify the next generation of SCN-resistant cultivars.

## Methods

### Plant populations development

Three recombinant inbred populations composed of 144, 131, and 244 $F_{3:4}$ lines were developed from crosses between PI 90763 × Peking, Forrest × PI 437654, and SA10-8471 × PI 90763, respectively, and population development was similar to the populations described previously[12]. All cross-hybridizations were made at the Bay Farm Research Facility in Columbia, Missouri during the summer of 2019 and populations were advanced using single-seed descent at winter nurseries in Hawaii and/or Puerto Rico before composite line establishment in later filial generations.

### SCN bioassay and population statistics

The soybean cyst nematode (SCN; *Heterodera glycines* Ichinohe) inbred populations PA3 (HG type 7/Race 3), TN7 (HG type 2.5.7/Race 1), MM4 (HG type 2.5.7/Race 5), and TN22 (HG type 1.2.5.7/Race 2) were used in this study[49,50]. Importantly, average female indices on the virulent HG type 1.2.5.7 indicator and race differential lines were as follows; Pickett (FI = 80), Peking (FI = 23), PI 88788 (FI = 92), PI 90763 (FI = 0), PI 437654 (FI = 0), PI 209332 (FI = 100), PI 89772 (FI = 0), and PI 548316 (FI = 79). SCN resistance bioassays were conducted in a greenhouse following established procedures in accordance with the Standardized Cyst Evaluation 2008 Protocol[51]. Seedlings from each line, along with parental lines, lines for HG type and race tests, and the susceptible control Williams 82 were transplanted into pots (100 cm³) of steam-pasteurized sandy loam soil. Each soybean line had five replicates and was organized in a randomized complete block design. Two days post transplanting, each seedling was inoculated with 1000 eggs by making a 6-cm depth hole at the base of the plant and dispensing inoculum into the hole. Pots were suspended in temperature-controlled water tanks to maintain a stable temperature of 27 °C throughout the experiment. Twenty-eight days post inoculation, each root system was soaked in water to remove soil and the females were collected by rinsing the root with high-pressure water over stacked 860 μm (no. 20) and 250 μm (no. 60) mesh sieves. The females from each sample were manually counted using a stereo microscope and the mean number from each line was obtained. Female index values were determined for each line by dividing the mean number of females from the test line by the mean number of females from the susceptible control line and multiplying by 100. Lines were rated in accordance with a standardized method as highly resistant (R, FI < 10), moderately resistant (MR, FI = 10–30), moderately susceptible (MS, FI = 31-60) and susceptible (S, FI > 60)[52]. Each plant population was phenotyped in separate tests. Shapiro-Wilk's test was performed in RStudio to determine the normality of the distribution of FI while symmetry was analyzed using through Skewness and Kurtosis of the distributions[53,54].

### DNA extraction and genotyping

A modified cetyl trimethyl ammonium bromide (CTAB) method was used for high-quality total genomic DNA extraction[55]. From each $F_{3:4}$ line, up to ten young trifoliate leaves were bulked. DNA concentrations were normalized to 100 ng/uL using a Mantis® automated liquid handler (Formulatrix). DNA samples extracted from all populations were submitted to the Soybean Genomics and Improvement Laboratory, USDA-ARS, for genotyping using the Illumina Infinium BARC-SoySNP6K BeadChip[56]. In addition to BARCSoySNP6K BeadChip genotyping, Kompetitive allele-specific PCR (KASP) assays were performed using four available markers for SCN resistance loci: (1) Rhg1-2[57] and (2) SNAP18-1[22] to differentiate *rhg1-a* allele, *rhg1-b* allele, and *Rhg1-c* susceptible allele; (3) SNAP11-1[22] to detect resistance at *rhg2*; and (4) Rhg4-5 to detect resistance at *Rhg4* locus[57]. The KASP genotyping procedure was performed based on standard protocol (LGC Genomics).

### Linkage mapping and fine-mapping analysis

Single-nucleotide polymorphism data obtained from Illumina Infinium BARCSoySNP6K BeadChip for both populations was filtered using TASSEL 5.0 software[58]. The minimum proportion of missing calls and maximum heterozygosity were determined at 90% and 30%, respectively. The names of SNPs were converted to their appropriate physical positions based on Wm82.a2.v1. The matrix was then converted to ABH format in TASSEL. The R package 'ABHgenotypeR'[59] was used to conduct imputation of the missing genotypes based on flanking alleles. To check the quality of association between the parents and developed lines, a similarity test was conducted using the R package 'ParentOff-Spring' with a threshold of 90%[60]. Mapping was performed using MapQTL 6.0 software[61]. Permutation tests were conducted in analyzed lines 1000 times, and initial logarithm of odds (LOD) threshold was established under type I error at alpha=0.05. Interval mapping (IM) at 1-cM intervals along the chromosomes was used to detect QTL based on initial LOD threshold of 3.0. Markers closely linked to positions with the highest LOD scores were taken as cofactors for multiple-QTL modeling (MQM) analysis. Graphical presentation of detected QTLs was drawn using MapChart 2.32 software[62].

After the initial linkage mapping and QTL discovery, we investigated the region of QTL02 in populations PI 90763 × Peking and SA10-8471 × PI 90763, as well as in two previously described populations, SA13-1385 × PI 90763 and LD11-2170 × PI 90763[12]. Among four populations, all lines that carried homozygous *GmSNAP18-a*, *GmSNAP11*, and *GmSHMT08* but not QTL12 were analyzed for differences in polymorphic markers as well as recombination sites. Nine $F_{3:4}$ lines derived from a cross PI 90763 × Peking that carried heterozygous region around QTL02, and homozygous *GmSNAP18-a*, *GmSNAP11*, and *GmSHMT08* were used to create $F_{4:5}$ sister lines with various recombination spots derived from each parent. Moreover, we advanced seven $F_{3:4}$ lines homozygous for *GmSNAP18-a*, *GmSNAP11*, *GmSHMT08* and the region of QTL02. Lines that could potentially carry QTL12 were not involved in fine-mapping. Primers for fourteen assays (MU-Gm02-01 through MU-Gm02-16) were developed using LGC Genomics KASP by design. All $F_{4:5}$ lines were genotyped with these assays and recombination spots were established. All developed $F_{4:5}$ lines were inoculated with TN22 population as described above. The score between resistance and moderate resistance was estimated based on standardized female index classification system as highly resistant (R, FI < 10) or moderately resistant (MR, FI = 10–30)[52].

### Whole-genome resequencing data analysis

We analyzed different haplotypes of *GmSNAP18, GmSNAP11* and *GmSNAP02* using the Soybean Allele Catalog Tool on SoyKB[24]. Raw sequencing reads of the *GmSNAP02* gene (*Glyma.02G260400*) were obtained from NCBI SRA (https://www.ncbi.nlm.nih.gov/sra) using the fastq-dump command in sra-tools v2.10.0, with the parameters "--gzip --origfmt --split-files". Read quality was assessed using FastQC v0.11.9 with default parameters and low-quality reads and Illumina adapter sequences were trimmed using Trimmomatic v0.39 with the parameters "ILLUMINACLIP:TruSeq3-PE.fa:2:30:10:2:True, Leading:3, Trailing:3, MINLEN:36". Quality-trimmed reads were aligned to the Williams 82 reference genome (Wm82.a2.v1 from Phytozome - https://phytozome-next.jgi.doe.gov/) using the bwa mem command in BWA v0.7.17 with default parameters. Mate coordinates and size fields were filled using the fixmate command in Samtools v1.13 with parameter "-m" to add mate score tags. Reads were coordinate sorted using samtools sort and duplicate reads marked with samtools markdup. Reads aligned to the region on Chr. 2 containing the *GmSNAP02* gene were extracted using samtools view command and visually inspected in the JBrowse 2 v1.7.7 desktop genome browser. Variant calling on all accessions were performed using the GATK v4.1.9.0 platform as previously described[63].

## Detection of a large insertion in *GmSNAP02*

Long-range PCR was performed to determine the size of the insertion of the *GmSNAP02-ins* haplotype. A set of primers F1/R1 (Supplementary Fig. 12) was developed to target the sequence flanking the insertion in the *GmSNAP02* gene. PCR reactions were carried out in 50 mL volumes containing final concentrations of 1× LongAmp *Taq* Buffer, 300 μM dNTPs, 0.5 μM of each primer, 5 units LongAmp Taq DNA Polymerase, and 80 ng DNA template. PCR was conducted on a MJ Research PTC-225 DNA Engine Tetrad Thermal Gradient Cycler using the following conditions: 94 °C for 5 min, followed by 30 cycles of 94 °C for 20 s, 55 °C for 45 s, 65 °C for 10 min. The resulting amplicons (20 μL) were visualized by running on a 0.8% agarose gel stained with Ethidium Bromide (Invitrogen) at 90 V for 1.5 h. 20 mL of a 1 kb Plus DNA ladder (Invitrogen) was included as a reference for molecular size. Long-range PCR products from PI 90763 were purified with QIAquick PCR & Gel Cleanup Kit (Qiagen) and run on a gel for quantification.

Forward and reverse primers F2/R2 (Supplementary Fig. 12) designed to the untranslated sequence immediately upstream of the start and stop codons for *GmSNAP02* were used to amplify the corresponding transcripts from Peking, PI 90763, and PI 437654 cDNA using Phusion® High-Fidelity DNA Polymerase (NEB) in a BioRad C1000 touch thermocycler with the following cycling parameters: 30 s at 98 °C, 40 cycles of 10 s at 98 °C, 30 s at 60 °C, 1.5 min at 72 °C followed by 7.5 min at 72 °C. The PCR products were separated by electrophoresis on a 1% agarose gel and purified using the QIAquick® gel extraction kit (Qiagen) following the manufacturer's protocol. For sequencing, purified fragments were cloned into the pGEM®- T Easy Vector (Promega).

## *GmSNAP02* expression analysis

Nematode infection of soybean seedlings was performed according to Ithal et al.[64]. Soybean seeds were germinated in ragdolls for three days at 28 °C in the dark. Seedlings with uniform radicles were selected for inoculation. SCN cysts were isolated from infested soil by collection on a 250 μm (no. 60) sieve and gently crushed using a drill press. Eggs were collected on a 25 μm (no. 500) sieve, sterilized in 0.02% sodium azide, and hatched on the antibiotic solution at 28 °C for 3 days. Approximately 300 J2s were applied 1 cm above the root tip of each soybean root. Mock-inoculated samples were treated the same except for the addition of nematodes. Inoculated soybean seedlings were kept at 26 °C. The infection was synchronized by washing inoculated roots after 24 h with running tap water to remove any J2s remaining outside the root. Washed seedlings were rolled into ragdolls, placed in Hoagland's nutrient solution with constant aeration, and placed in a plant growth chamber at 26 °C with a photoperiod of 16 h of light and 8 h of dark for an additional two days. Three days post inoculation (dpi), root pieces of ~1–1.5 cm flanking the inoculation point were excised from both inoculated and mock-inoculated roots. Excised root pieces were immediately flash-frozen in liquid nitrogen and stored at −80 °C until RNA extraction. Three biological replicates consisting of 10–12 seedlings/treatment were used for each genotype. The nematode infection process in Peking, PI 90763, and PI 437654 at 3 and 5 dpi was monitored by staining with acid fuchsin[65].

## RNA extraction and real-time qRT-PCR analysis

Total RNA was isolated from soybean root samples using the RNeasy Plant Mini kit (Qiagen) following the manufacturer's protocol. Total RNA was used to synthesize cDNA using the PrimeScript 1st strand cDNA synthesis kit (Takara). Real-time quantitative reverse transcription polymerase chain reaction (qRT-PCR) analysis was carried out using PowerUp™ SYBR™ green master mix (Applied Biosystems) in a CFX96 C1000 touch thermal cycler (Bio-rad). The gene-specific primer sequences used in qRT-PCR analysis are listed in Supplementary File S12. The expression level of each gene tested was normalized to two soybean reference genes *GmUbiquitin* (accession no. D28123)[66] and *GmTUA5* (accession no. AY907702)[67]. Relative expression was calculated by the Pfaffl method[68].

## *GmSNAP02* CRISPR guide RNA (gRNA) and construct design

A dual sgRNA plasmid construction system was used as described in ref. 69. The CRISPR/Cas9 construct backbone 35S-Cas9-SK and sgRNA template plasmid AtU6-26-SK were gifts from Jiam-Kang Zhu's lab. This system utilizes the *Arabidopsis AtU6-26* promotor to drive expression of the human codon-optimized *Streptococcus pyogenes* Cas9 (hspCas9)[25,70]. Several *GmSNAP02* gRNA sequences for CRISPR/Cas9 targeting different regions of *GmSNAP02* were designed using the CHOPCHOP web tool[71]. Complementary oligomers for selected gRNA were designed and annealed double-stranded gRNA sequences were cloned into the AtU6-26-SK vector. The *Cas9* cassette from 35S-Cas9-SK and the sgRNA expression cassettes were subcloned into the pcamGFP-CvMV-GWOX binary vector. The pcamGFP-CvMV-GWOX binary vector has a strong *CvMV* promoter driving a *GFP* reporter gene cassette for transgene selection. The final constructs were introduced into the *Rhizobium rhizogenes* strain K599 using the freeze-thaw method[72]. Two different pcamGFP-CvMV-*GmSNAP02*-gRNAa-gRNAb constructs confirmed for *GmSNAP02* edits by sequencing were used to generate soybean composite plants with transgenic roots. The final constructs were named as pcamGFP-CvMV-*GmSNAP02*-T3-T4 and pcamGFP-CvMV-*GmSNAP02*-T5-T7. The pcamGFP-CvMV-GWOX construct with no gRNA was used as the empty vector (EV) control. Primers used for vector construction are in the Supplementary Fig. 12.

## Generation of composite soybean plants

Composite soybean plants with *GmSNAP02*-edited roots were generated by *Rhizobium rhizogenes* (K599) mediated transformation according to the method of Fan et al.[73] with modifications for SCN bioassays. Following K599 inoculation, plants were sealed and kept in a growth chamber set to 26 °C with a photoperiod of 16 h light/8 h dark for hairy root generation before moving to the greenhouse. Composite plants with transformed hairy roots were selected under fluorescent light and untransformed roots were removed. Plants with uniform roots were transplanted to 1:1 sand:soil mix for SCN bioassays. Two independent biological replicates, each containing 14 plants for two different *GmSNAP02* knockout events and EV were used for phenotyping. The first and second replicates of soybean composite plants were inoculated with 3000 and 2000 SCN HG type 1.2.5.7 eggs, respectively. Inoculated plants were kept in the greenhouse at 26 °C with a 16 h/8 h light/dark period for 28 days. Cysts from roots were extracted and counted using a stereoscope. A Kruskal–Wallis statistical test was used to determine significant differences in cyst counts in between Peking plants transformed with different CRISPR/Cas9-gRNAa+gRNAb constructs. Since there was a significant difference, we carried out a pair-wise comparison using the Wilcoxon rank sums test. The dot plot was created using the R program ver. 4.2.2.

## CRISPR/Cas9 deletion check and off-target analysis

Genomic DNA was extracted from selected transformed roots of both PI 90763 and Peking using the CTAB-chloroform-based method[74]. To confirm CRISPR/Cas9 edits, PCR amplification was conducted using *GmSNAP02*-specific primers flanking target sites and Phusion® High-Fidelity DNA Polymerase (NEB) in a BioRad C1000 touch thermocycler with the following cycling parameters: 30 s at 98 °C, 40 cycles of 10 s at 98 °C, 30 s at 60 °C, 1.5 min at 72 °C followed by 7.5 min at 72 °C. PCR amplicons were gel extracted and sequenced directly. SNAPGene viewer was used to decode the chromatograms. If the deletion/insertion was not clear in chromatograms, the PCR amplicons were A-tailed and cloned to pGEM-T Easy (Promega), and Sanger sequenced to confirm deletions. Potential off-target sites with 3 bp or fewer than 3 bp mismatches in the 12-bp seed sequence of the four sgRNA of *GmSNAP02* predicted by the CHOPCHOP online tool were amplified

using sequence-specific primers (Supplementary Fig. 12) PCR products amplified from WT and edited plants were sequenced.

## Specific allele combination evaluations

$F_{3:5}$ lines with various homozygous allele combinations were pooled from four populations PI 90763 × Peking, SA10-8471 × PI 90763, SA13-1385 × PI 90763, and LD11-2170 × PI 90763. The lines were confirmed using available KASP assays for *GmSNAP18*, *GmSNAP11*, and *GmSHMT08*[22,57], and closest polymorphic SNPs to the *GmSNAP02* gene based on BARCSoySNP6K matrix. The lines were phenotyped against four SCN populations: TN7 (HG type 2.5.7/Race 1), TN22 (HG type 1.2.5.7/Race 2), PA3 (HG type 7/Race 3), and MM4 (HG type 2.5.7/Race 5). The SCN bioassay was described above. Specific allele combinations indicate homozygous resistance alleles: *GmSNAP18-a* and *GmSNAP18-b* correspond to *rhg1-a* and *rhg1-b* alleles; *GmSNAP11* corresponds to *rhg2*, *GmSHMT08* corresponds to *Rhg4*, and *GmSNAP02-ins* corresponds to the allele with a large insertion in exon 8 of *GmSNAP02*. Female indices derived from lines infected from each SCN population were analyzed by one-way analysis of variance (ANOVA) using agricolae package in RStudio, and Tukey's HSD test was used for multiple comparisons at $P \leq 0.05$[75]. Violin plots were created in R using ggplot2 package for the data visualization[76].

## Molecular marker development

Mismatched short raw sequencing reads aligned to the exon 8 of the *GmSNAP02* gene were extracted from the JBrowse 2 v1.7.7 desktop genome browser. PrimerQuest™ Tool was used to design two TaqMan primer pairs and two probes for the *SNAP02* assay, which was encoded as MU-SNAP02[INS]-WT and MU-SNAP02[INS]-MUT in our marker library. Assay MU-SNAP02[INS]-WT was developed to amplify and detect an amplicon of 121 bp from the wild-type *GmSNAP02* allele like in Williams 82 and Peking. The second assay MU-SNAP02[INS]-MUT was developed to detect an amplicon of 133 bp in mutant soybean lines which included the 3'end of the insertion. TaqMan assays MU-SNAP02[DEL]-1 and MU-SNAP02[DEL]-2 were developed to detect the *GmSNAP02-del* haplotype. At the position of Gm02: 44,695,753 (Wm82.a2.), three polymorphic variants are present: (1) REF like Williams 82 and Peking carries a reference genome sequence "CAGGGCCGAGGGATTTTGAGAACA"; (2) ALT like Forrest (PI 548655) and PI 88788 carries an alternate sequence "GAGGGCCGAGGATTTTGAGAACA"; and (3) DEL like PI 437654 and PI 89772 carries the 22 nucleotides deletion sequence "C" resulting in frameshift at the eight amino acid (*GmSNAP02-del* haplotype).

## Reporting summary

Further information on research design is available in the Nature Portfolio Reporting Summary linked to this article.

## Data availability

The authors declare that the data supporting the findings of this study are available within the paper, within the supplementary information file, and within the data source file. The raw whole-genome sequencing data used within this paper are available for the four accessions as follows; PI 90763 [https://www.ncbi.nlm.nih.gov/sra/?term=SRR2163296], PI 437654 [https://www.ncbi.nlm.nih.gov/sra/?term=SRR2163307], Peking [https://www.ncbi.nlm.nih.gov/sra/?term=SRR2163294], and Williams 82 [https://ngdc.cncb.ac.cn/gsa/browse/CRA002269/CRR108703]. Plant material and soybean cyst nematode populations used in this manuscript are available upon request via a material transfer agreement and associated permits with each respective institution. Source data are provided with this paper.

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

## Acknowledgements

Special thanks to Kurk Lance and Dean Kemp at the University of Georgia for technical assistance with SCN population maintenance and bioassays. The authors acknowledge financial support of this research from the Missouri Soybean Merchandising Council (11-340-20–23; 16-400-20–23; and 16-398-20–23 to A.M.S.), the United Soybean Board (2020-172-0152, 2120-172-0143, 2220-172-0151, 2312-209-0301 to A.M.S. and M.G.M.), and the National Science Foundation and U.S. Department of Agriculture National Institute of Food and Agriculture Plant-Biotic Interactions Program (Grant/Award 2021-67013-35887 to M.G.M.). V.A.G. was funded in part by an Institute of Plant Breeding, Genetics, and Genomics Graduate Research Fellowship.

## Author contributions

All authors (M.U., V.A.G., C.G.M., N.D., M.T., P.B., J.D.G., K.D.B., Q.S., B.D., A.N., M.G.M., and A.M.S.) contributed to the conception and design of the article, interpretation of the relevant literature, and revision of the manuscript. M.U. conceived the study, developed mapping populations, performed QTL mapping and fine-mapping, analyzed haplotypes, and wrote the manuscript draft. V.A.G. analyzed haplotypes, conducted SCN infection and expression studies, designed CRISPR-Cas9 constructs, phenotyped composite plants, and wrote the manuscript draft. C.G.M., B.D., and A.N. phenotyped all plant material for mapping studies. N.D. and K.D.B. analyzed raw WGRS reads. M.T. was involved in QTL mapping. P.B. developed mapping populations and performed statistical analyses. J.D.G. performed QTL mapping analysis. Q.S. genotyped mapping populations. M.G.M and A.M.S. conceived the study, acquired funding, supervised the project, and helped write the manuscript. All authors have read and agreed to the published version of the manuscript.

## Competing interests

The authors have filed the following US provisional patent applications—No. 63/387,635 filed on 12/15/2022 and No. 63/503,811 filed on 05/23/2023.
