## [Peer Review File · Nature Communications]

Loss-of-Function of an α -SNAP Gene Confers Resistance to Soybean Cyst NematodeREVIEWER COMMENTS

Reviewer #1 (Remarks to the Author):

This study identified a novel SNAP gene, GmSNAP02, through gene mapping using three RIL populations, and validated the candidate gene using CRISPR-Cas9 genome editing. The manuscript was well organized. However, the weakness of this study is the novelty. The mechanism of SNAP in SCN-resistance is well studied, and gain of resistance through dysfunction of SNAP is anticipated according to its molecular mechanism in SCN-resistance. Moreover, how to exclude the other 33 candidate genes identified through fine mapping? For the CRISPR-Cas9 genome editing experiment, I noticed transient transgenic hairy roots were used. To my knowledge, the gRNA-induced gene editing is difficult to be homozygous in T0 generation, even in the hairy roots. I wonder how to obtain homozygous transgenic hairy roots for gene editing in your study? Therefore, owing to the above major concerns, I don't recommend to publish the manuscript on the journal of NC.

Reviewer #2 (Remarks to the Author):

Review of Usovsky et al., Loss-of-Function of a Novel Soluble N-ethylmaleimide-Sensitive Factor Attachment Protein Gene Confers Resistance to Soybean Cyst Nematode

In this manuscript, the authors define a new resistance locus for soybean cyst nematode (SCN) which is a member of the SNAP gene family, a gene family where two other members (on Chromosomes 11 and 18) have been previously shown to confer SCN resistance. The authors map the locus first via QTL analysis and then via fine mapping in a biparental cross, identify the causative gene, show its expression responds to SCN and validate its activity via genome editing.

Interestingly, though, while for example the chromosome 18 alpha-SNAP requires overexpression as part of a large tandem repeat to confer resistance, in this case the absence of the functional gene (via natural or artificial means) confers resistance. The experiments that map the SCN resistance trait to the Chromosome 2 locus, fine map it to a genomic region and replicate the SCN resistance phenotype using CRISPR/Cas9 genome editing are carefully done and completely convincing. The phenotype observed, especially convincing in the isogenic conditions of the genome edited roots, is surprisingly strong, showing that this locus is very likely of high economic value in soybean cultivation. SCN is the biggest single cause of yield loss in soybean, a crop with a global value of over \$150bn and a key source of protein and oil for the world's population. The only means of control of SCN is via genetic resistance and current resistance alleles are being overcome, so this work is timely and of extremely high impact.

I have one or two minor questions of this work as it stands currently, however these do not affect the main conclusions:

1. Is the inheritance of resistance at GmSNAP02 recessive, or partially or completely dominant? The loss of function hypothesis would suggest that the resistance gene should be completely or at least mostly recessive, recessive resistance genes are rare so this is worth reporting.

2. The two control genes used for the qPCR are ubiquitin and tubulin. Since the ubiquitin system is tightly integrated with the membrane trafficking system, it would be reassuring to see that the result is comparable when normalization to both of the genes is done separately. At the moment it appears that the result is normalized to an average of both genes.

3. The strategy of pyramiding these resistance loci that is suggested at several points in the MS is subject to two caveats: a) this gene cannot have a substantial yield penalty, or the economics of using it would offset any resistance advantage, and b) the advantage of pyramiding would be offset by genetic interactions between the loci, such as epistasis.

a) Do the authors have any evidence of an association between this Chromosome 2 locus and yield, in either direction?

b) Since epistatic interactions were observed both between the major QTL on chromosomes 2 and 11 in this work, and between 11 and 18 as previously described, as well as between Rhg1 and Rhg4, the allele stacking may not increase resistance. The directions and extents of these interactions are worth further consideration.

4. This SNAP presumably acts via interaction with the downstream SNARE proteins previously identified. It would be interesting to see how this SNAP interacts with the SNARE proteins downstream of the Rhg1 Chr18 SNAP, and the NSF RAN07, since its effects seem to be in the opposite direction (i.e. knockout confers rather than eliminates resistance).

Reviewer #3 (Remarks to the Author):

I am really impressed with this paper. What is presented is a clear account of the research, with really excellent attention to detail (e.g. the variant analysis and the phenotyping of 3dpi vs 5dpi infection before qPCR). Ultimately, the paper presents the discovery of a new nematode S gene. More than just discovery, it shows implementation of genome editing in a crop to provide resistance to one of the most affected crops by nematodes world-wide. Truly exceptional. S genes are going to be the future of crop protection in many parts of the world because of CRISPR. In addition to the importance, the discovery itself is very interesting. Another SNAP (!), but this time a loss of function(S gene) required in combination with specific sets of alleles to be functional was stunning (and explains why it was not found before). Taken together, this is a landmark paper in the field and I recommend it is accepted for publication following the most minor of revisions detailed below.

Specific suggestions:

GmSNAP02 is clearly an S gene that is dependent on genetic background. This is very interesting. It would be good to introduce the idea of S genes earlier on (probably introduction), compare this S gene to others known (ideally for nematodes, although there aren't too many known), and compare more broadly (are S genes normally "context dependent" as this one is?).

The following sentence reads that forward genetics was not used, which of course was the bulk of this paper (forward genetics mapping experiments often use reverse genetics at some point to confirm the gene, but fundamentally the approach was forward genetics). Reverse genetics can and has been used to identify S genes against nematodes, but this is not that.

“To date, these so-called susceptibility genes (S-genes) have been primarily identified through forward genetic studies. Here, we used gene editing as a reverse genetic tool for targeted knock out of the GmSNAP02 gene in the cultivar Peking, confirming this gene as a new and vitally important S-gene in soybean.”

Given an S gene, we normally expect the resistance to be recessive. Three related questions here: 1) Does that fit with the early genetics (segregating populations); Does it fit with the lines from the four mapping populations derived from crosses with PI 90763 as one of the parents; and does it fit with the CRISPR (were the edits bi-allelic?).

Response to the Reviewers' Comments

Comments from Reviewer #1

1. This study identified a novel SNAP gene, GmSNAP02, through gene mapping using three RIL populations, and validated the candidate gene using CRISPR-Cas9 genome editing. The manuscript was well organized. However, the weakness of this study is the novelty. The mechanism of SNAP in SCN-resistance is well studied, and gain of resistance through dysfunction of SNAP is anticipated according to its molecular mechanism in SCN-resistance.

Response from authors – From a historical standpoint, α -SNAPs represent a ‘recently’ discovered novel, non-canonical class of resistance proteins used by plants to defend against nematodes. Although α -SNAP GmSNAP18 dysfunction is thought to contribute to SCN resistance through the disruption of cellular vesicular trafficking pathways, the underlying molecular mechanism and full spectrum of molecular players (both plant and nematode) remains unknown. Thus, we believe our findings presented in this manuscript are unprecedented for several reasons. First, this study identifies a new molecular player, GmSNAP02. GmSNAP02’s identity and role in resistance was previously unknown, only discovered through the use of a novel biparental crossing strategy coupled with CRISPR-Cas9 gene editing. We not only provide the first evidence that GmSNAP02 plays a role in resistance to SCN but we conclusively demonstrate that the mechanism of action is through it is a loss-of-function of GmSNAP02. GmSNAP02 is the first soybean gene identified to play a role in resistance to SCN through a loss-of-gene function contributing to resistance was unexpected and surprising, and has revealed a potential nematode virulence target in the host. For these reasons, we believe our results represent a major breakthrough in the field that will not only have immediate impact in the mechanistic understanding of the plant’s resistance against this parasite but will also shed new light on how the nematode may be targeting a key protein and conserved cellular process to not only cause disease on soybean, but defeat native resistance genes. Our findings add a critical new piece to the puzzle as we advance our limited understanding of this type of pathogen resistance. Moreover, α -SNAP proteins represent key housekeeping proteins involved in membrane trafficking processes largely conserved across eukaryotes underscoring the far-reaching implications of the work.

2. Moreover, how to exclude the other 33 candidate genes identified through fine mapping?

Response from authors – The authors evaluated all candidate genes in the interval and prioritized them based on functional annotations. The GmSNAP02 gene, encoding a protein belonging to the α -SNAP family, was identified as a top candidate for the reasons described above. In addition, we demonstrated that GmSNAP02 was upregulated in response to SCN infection in Peking (S), but not PI 90763 (R) (Figure 3b). CRISPR-Cas9 edited Peking root systems confirmed to have the full 5’ end deletion (T4+T3) in GmSNAP02 reverted to a strong SCN resistant phenotype (e.g., Peking #1, no cysts = 1; Figure 4d,e). These results, coupled with the identification of GmSNAP02 loss-of-function mutations in resistant soybean lines, we unequivocally demonstrated GmSNAP02 is the major contributing gene to SCN resistance at this locus. Additionally, a full list of all 34 genes within the fine mapped interval are available in the supplementary file.

3. For the CRISPR-Cas9 genome editing experiment, I noticed transient transgenic hairy roots were used. To my knowledge, the gRNA-induced gene editing is difficult to be homozygous in T0 generation, even in the hairy roots. I wonder how to obtain homozygous transgenic hairy roots for gene editing in your study?

Response from authors – Yes, we employed a soybean composite plant system (wild-type shoots with transgenic hairy roots) for our CRISPR-Cas9 genome editing experiments to unequivocally demonstrate GmSNAP02 contributes to SCN resistance via loss-of-function in Peking. Using a dual guide system, we were able to generate homozygous deletions in the T0 generation as demonstrated by PCR and sequencing (Figure 4d,e; e.g., Peking #1, cyst no. = 1). In some cases, we did not observe the full deletion by PCR, but sequencing revealed small indel edits in GmSNAP02 and reversion to the SCN resistant phenotype in these plants (Figure 4d,e; e.g., Peking #5, cyst no. 2) indicating homozygous edits in these plants as well. Roots exhibiting no GmSNAP02 edits by PCR and sequencing exhibited a susceptible SCN phenotype (Figure 4d,e; e.g. Peking #3 and #6, cyst no. 114 and 65). Although we cannot conclude with certainty the mode of GmSNAP02 inheritance from our CRISPR and mapping studies (see below), homozygous edits may not be necessary to detect the SCN resistance phenotype.

Comments from Reviewer #2

1. Review of Usovsky et al., Loss-of-Function of a Novel Soluble N-ethylmaleimide-Sensitive Factor Attachment Protein Gene Confers Resistance to Soybean Cyst Nematode. In this manuscript, the authors define a new resistance locus for soybean cyst nematode (SCN) which is a member of the SNAP gene family, a gene family where two other members (on Chromosomes 11 and 18) have been previously shown to confer SCN resistance. The authors map the locus first via QTL analysis and then via fine mapping in a biparental cross, identify the causative gene, show its expression responds to SCN and validate its activity via genome editing. Interestingly, though, while for example the chromosome 18 alpha-SNAP requires overexpression as part of a large tandem repeat to confer resistance, in this case the absence of the functional gene (via natural or artificial means) confers resistance. The experiments that map the SCN resistance trait to the Chromosome 2 locus, fine map it to a genomic region and replicate the SCN resistance phenotype using CRISPR/Cas9 genome editing are carefully done and completely convincing. The phenotype observed, especially convincing in the isogenic conditions of the genome edited roots, is surprisingly strong, showing that this locus is very likely of high economic value in soybean cultivation. SCN is the biggest single cause of yield loss in soybean, a crop with a global value of over \$150bn and a key source of protein and oil for the world's population. The only means of control of SCN is via genetic resistance and current resistance alleles are being overcome, so this work is timely and of extremely high impact. I have one or two minor questions of this work as it stands currently, however these do not affect the main conclusions:

Is the inheritance of resistance at GmSNAP02 recessive, or partially or completely dominant? The loss of function hypothesis would suggest that the resistance gene should be completely or at least mostly recessive, recessive resistance genes are rare so this is worth reporting.

Response from authors – The authors make note here that it is extremely important to answer this question with the appropriate experimental design, population development and structure, and subsequent experimental protocols as to not falsely claim GmSNAP02 follows a complete dominance

genetic effect, where the recessive alleles confer resistance as suggested. To evaluate the genetic effect of GmSNAP02 precisely and accurately (i.e., complete dominance, additive, epistatic, incomplete dominance, etc.) heterozygous individuals need to be evaluated and subsequently defined to a specific phenotypic range within a large F₂ segregating population(s), due to the known additional genes which also impart SCN resistance in an epistatic manner. For our population development method to implement QTL mapping and fine mapping, we utilized recombinant inbred lines (RIL) derived in the F₃ generation via single-seed descent with populations no larger than approximately 300 individuals. We then evaluated 5 plants representing 5 replications of each RIL to determine female index for each line. Molecular genotype data (SNP's) are then quality controlled by removing heterozygous loci as well as markers that do not follow expected segregation ratios to reduce type III error. While these methods and experimental design are completely appropriate for genetic mapping studies to elucidate molecular marker associations and QTL intervals, they are not appropriate for an ad hoc analysis of the genetic effect of a single gene, particularly because the difference between a complete dominance effect and an incomplete dominance effect cannot be readily distinguished.

2. The two control genes used for the qPCR are ubiquitin and tubulin. Since the ubiquitin system is tightly integrated with the membrane trafficking system, it would be reassuring to see that the result is comparable when normalization to both of the genes is done separately. At the moment it appears that the result is normalized to an average of both genes.

Response from authors – To reassure the reviewers concerns the authors normalized both genes separately. Both reference genes exhibit consistent expression levels between mock and TN22-infected root tissues across all genotypes. Moreover, both genes demonstrated relatively higher expression, with only minor differences in expression (based on Cq values). Therefore, averaging the expression of these two reference genes had a negligible impact on normalization. Notably, when normalized to the reference genes individually, GmSNAP02 showed an identical expression pattern. We have provided the attached graphs to provide a clear illustration of these findings.

3. The strategy of pyramiding these resistance loci that is suggested at several points in the MS is subject to two caveats: a) this gene cannot have a substantial yield penalty, or the economics of using it would offset any resistance advantage, and b) the advantage of pyramiding would be offset by genetic interactions between the loci, such as epistasis.

Response from authors – The authors generally agree with this statement, but we currently do not have the appropriate data to confirm or deny the hypothesized/potential yield penalty, and we present detailed figures and data highlighting the effect of pyramiding GmSNAP02 with other known SCN genes and alleles in Figure 5 and in supplementary figures.

4. Do the authors have any evidence of an association between this Chromosome 2 locus and yield, in either direction?

Response from authors – The authors do not currently have preliminary data that illustrates the potential correlation between alleles of GmSNAP02 and grain yield potential, although we are currently developing BC₄ lines at the University of Missouri for direct testing of potential yield drag.

5. Since epistatic interactions were observed both between the major QTL on chromosomes 2 and 11 in this work, and between 11 and 18 as previously described, as well as between Rhg1 and

Rhg4, the allele stacking may not increase resistance. The directions and extents of these interactions are worth further consideration.

Response from authors – The authors demonstrate and illustrate the effect of the addition of the GmSNAP02 allele with Rhg2, Rhg1, and Rhg4 in Figure 5, as well as in the supplementary file figures. Since our initial discovery of GmSNAP02’s impact on SCN resistance, the optimal breeding strategy has been introduced to the soybean breeding program at University of Missouri. After three years of “breeding by design” and subsequent evaluation of selections using molecular markers, we can confirm that stacking the specific alleles GmSNAP02-ins and GmSNAP02-del works as efficiently as described in this manuscript.

6. This SNAP presumably acts via interaction with the downstream SNARE proteins previously identified. It would be interesting to see how this SNAP interacts with the SNARE proteins downstream of the Rhg1 Chr18 SNAP, and the NSF RAN07, since its effects seem to be in the opposite direction (i.e. knockout confers rather than eliminates resistance).

Response from authors – Yes, the authors agree that it will be very interesting to further explore the underlying molecular interactions of GmSNAP02 with the vesicular trafficking machinery to decipher how it contributes to resistance and this question is a focus of our future work.

Comments from Reviewer #3

1. I am really impressed with this paper. What is presented is a clear account of the research, with really excellent attention to detail (e.g. the variant analysis and the phenotyping of 3dpi vs 5dpi infection before qPCR). Ultimately, the paper presents the discovery of a new nematode S gene. More than just discovery, it shows implementation of genome editing in a crop to provide resistance to one of the most affected crops by nematodes world-wide. Truly exceptional. S genes are going to be the future of crop protection in many parts of the world because of CRISPR. In addition to the importance, the discovery itself is very interesting. Another SNAP (!), but this time a loss of function (S gene) required in combination with specific sets of alleles to be functional was stunning (and explains why it was not found before). Taken together, this is a landmark paper in the field and I recommend it is accepted for publication following the most minor of revisions detailed below.

Specific suggestions:

GmSNAP02 is clearly an S gene that is dependent on genetic background. This is very interesting. It would be good to introduce the idea of S genes earlier on (probably introduction), compare this S gene to others known (ideally for nematodes, although there aren’t too many known), and compare more broadly (are S genes normally “context dependent” as this one is?).

Response from authors – Yes, the authors mostly agree that GmSNAP02 appears to be behaving like an S gene, although in a nematode population and genetic background dependent manner, and therefore does not fit the classic definition of an S-gene. Thus, we chose not to introduce the concept of S-genes in the introduction, but rather allude to this interesting observation in the discussion. It remains to be demonstrated whether we can knock-out GmSNAP02 in a susceptible soybean

background (i.e., one that carries no known resistance alleles and is susceptible to all SCN populations). We have added some additional discussion in this regard to the manuscript.

2. The following sentence reads that forward genetics was not used, which of course was the bulk of this paper (forward genetics mapping experiments often use reverse genetics at some point to confirm the gene, but fundamentally the approach was forward genetics). Reverse genetics can and has been used to identify S genes against nematodes, but this is not that.
“To date, these so-called susceptibility genes (S-genes) have been primarily identified through forward genetic studies. Here, we used gene editing as a reverse genetic tool for targeted knock out of the GmSNAP02 gene in the cultivar Peking, confirming this gene as a new and vitally important S-gene in soybean.”

Response from authors – The authors have updated this sentence and paragraph for clarity in the revised manuscript.

3. Given an S gene, we normally expect the resistance to be recessive. Three related questions here: 1) Does that fit with the early genetics (segregating populations)

Response from authors – The authors agree with this hypothesis, in general, although we make note here that it is extremely important to answer this question with the appropriate experimental design, population development and structure, and subsequent experimental protocols as to not falsely claim GmSNAP02 follows a complete dominance genetic effect, where the recessive alleles confer resistance as suggested. To evaluate the genetic effect of GmSNAP02 precisely and accurately (i.e., complete dominance, additive, epistatic, incomplete dominance, etc.) heterozygous individuals need to be evaluated and subsequently defined to a specific phenotypic range within a large F_2 segregating population(s), due to the known additional genes which also impart SCN resistance in an epistatic manner. For our population development method to implement QTL mapping and fine mapping, we utilized recombinant inbred lines (RIL) derived in the F_3 generation via single-seed descent with populations no larger than approximately 300 individuals. We then evaluated 5 plants representing 5 replications of each RIL to determine female index for each line. Molecular genotype data (SNP's) are then quality controlled by removing heterozygous loci as well as markers that do not follow expected segregation ratios to reduce type III error. While these methods and experimental design are completely appropriate for genetic mapping studies to elucidate molecular marker associations and QTL intervals, they are not appropriate for an ad hoc analysis of the genetic effect of a single gene, particularly because the difference between a complete dominance effect and an incomplete dominance effect cannot be readily distinguished.

4. Does it fit with the lines from the four mapping populations derived from crosses with PI 90763 as one of the parents, and

Response from authors – As above, the authors make note here that it is extremely important to answer this question with the appropriate experimental design, population development and structure, and subsequent experimental protocols as to not falsely claim GmSNAP02 follows a complete dominance genetic effect, where the recessive alleles confer resistance as suggested. To evaluate the genetic effect of GmSNAP02 precisely and accurately (i.e., complete dominance, additive,

epistatic, incomplete dominance, etc.) heterozygous individuals need to be evaluated and subsequently defined to a specific phenotypic range within a large F₂ segregating population(s), due to the known additional genes which also impart SCN resistance in an epistatic manner. For our population development method to implement QTL mapping and fine mapping, we utilized recombinant inbred lines (RIL) derived in the F₃ generation via single-seed descent with populations no larger than approximately 300 individuals. We then evaluated 5 plants representing 5 replications of each RIL to determine female index for each line. Molecular genotype data (SNP's) are then quality controlled by removing heterozygous loci as well as markers that do not follow expected segregation ratios to reduce type III error. While these methods and experimental design are completely appropriate for genetic mapping studies to elucidate molecular marker associations and QTL intervals, they are not appropriate for an ad hoc analysis of the genetic effect of a single gene, particularly because the difference between a complete dominance effect and an incomplete dominance effect cannot be readily distinguished.

5. does it fit with the CRISPR (were the edits bi-allelic?)

Response from authors – Yes, it does support a recessive or mostly recessive mode of inheritance. We employed a soybean composite plant system (wild-type shoots with transgenic hairy roots) for our CRISPR-Cas9 genome editing experiments to unequivocally demonstrate GmSNAP02 contributes to SCN resistance via loss-of-function in Peking. Using a dual guide system, we were able to generate homozygous deletions in the T0 generation as demonstrated by PCR and sequencing (Figure 4d,e; e.g., Peking #1, cyst no. = 1). In some cases, we did not observe the full deletion by PCR, but sequencing revealed small indel edits in GmSNAP02 and reversion to the SCN resistant phenotype in these plants (Figure 4d,e; e.g., Peking #5, cyst no. 2) indicating homozygous edits of these plants as well. Roots exhibiting no GmSNAP02 edits by PCR and sequencing exhibited a susceptible SCN phenotype (Figure 4d,e; e.g. Peking #3 and #6, cyst no. 114 and 65).

REVIEWERS' COMMENTS

Reviewer #2 (Remarks to the Author):

The authors have satisfied all of my concerns with a response that is notable for its attention to detail. I accept that although I am very interested to know whether this gene is inherited in a recessive manner, exactly what the inheritance pattern of the stacked alleles are, and whether these interactions can be characterized at the protein-protein level, that these results would be enough for a second manuscript. The results as presented are amply sufficient for a very high impact publication.

Note that I was unable to find the chart for the individual qRT-PCR control genes, although numeric data was provided in a strangely formatted table. I am prepared to take the authors word for the control genes behaving near-identically, but it looks like something went wrong here, probably in the conversion of an Excel file to PDF. If these are to be published, the PDF "430377_1_related_ms_8001808_s02dl4.pdf" needs to be reformatted in a way that is readable.

Reviewer #3 (Remarks to the Author):

I found the response to reviews lacking (generally so, but also specifically to my comments).

I strongly recommend the following two points are addressed:

1. GmSNAP02 is an S gene. I am not aware of a widely accepted view that S genes: a) don't depend on genetic background (why would they not?); nor b) don't depend on the pathogen population (why would they not?). As such, I find the response lacking, and it very unusual to not mention S genes in the introduction.

2. My understanding of the following sentences can be paraphrased to, "most people do forward genetics but we did reverse genetics" - which of course is factually incorrect: "To date, these so-called susceptibility genes (S-genes) have been primarily identified through forward genetic studies. Here, we used gene editing as a reverse genetic tool for targeted knock out of the GmSNAP02 gene in the cultivar Peking, confirming this gene as a new and vitally important S-gene in soybean."

To avoid misunderstanding, I suggest the following (or just remove the preceding sentence entirely), "To date, these so-called susceptibility genes (S-genes) have been primarily identified through forward genetic studies. Here, we also used forward genetics, followed by gene editing for targeted knock out of the GmSNAP02 gene in the cultivar Peking, confirming this gene as a new and vitally important S-gene in soybean."

Response to the Reviewers' Comments – 10/22/23

Reviewer #2 (Remarks to the Author):

The authors have satisfied all of my concerns with a response that is notable for its attention to detail. I accept that although I am very interested to know whether this gene is inherited in a recessive manner, exactly what the inheritance pattern of the stacked alleles are, and whether these interactions can be characterized at the protein-protein level, that these results would be enough for a second manuscript. The results as presented are amply sufficient for a very high impact publication.

Note that I was unable to find the chart for the individual qRT-PCR control genes, although numeric data was provided in a strangely formatted table. I am prepared to take the authors word for the control genes behaving near-identically, but it looks like something went wrong here, probably in the conversion of an Excel file to PDF. If these are to be published, the PDF "430377_1_related_ms_8001808_s02dl4.pdf" needs to be reformatted in a way that is readable.

Response from authors –

First, the authors would like to thank Reviewer #2 for their thoughtful insight and review of this manuscript, as well as their time and effort in the preparation thereof. The authors additionally apologize for the absence of the chart and formatted table due to conversion to .pdf during upload. All data can now be found in the Source Data file.

Reviewer #3 (Remarks to the Author):

I found the response to reviews lacking (generally so, but also specifically to my comments).

I strongly recommend the following two points are addressed:

1. GmSNAP02 is an S gene. I am not aware of a widely accepted view that S genes: a) don't depend on genetic background (why would they not?); nor b) don't depend on the pathogen population (why would they not?). As such, I find the response lacking, and it very unusual to not mention S genes in the introduction.

Response from authors –

First, the authors would like to thank Reviewer #3 for their thoughtful insight and review of this manuscript, as well as their time and effort in the preparation thereof. The authors have added the concept of S genes and their role in plant resistance to the introduction as requested.

2. My understanding of the following sentences can be paraphrased to, "most people do forward genetics but we did reverse genetics" - which of course is factually incorrect: "To date, these so-called susceptibility genes (S-genes) have been primarily identified through forward genetic studies. Here, we used gene editing as a reverse genetic tool for targeted knock

out of the GmSNAP02 gene in the cultivar Peking, confirming this gene as a new and vitally important S-gene in soybean.”

To avoid misunderstanding, I suggest the following (or just remove the preceding sentence entirely), "To date, these so-called susceptibility genes (S-genes) have been primarily identified through forward genetic studies. Here, we also used forward genetics, followed by gene editing for targeted knock out of the GmSNAP02 gene in the cultivar Peking, confirming this gene as a new and vitally important S-gene in soybean.”

Response from authors –

Thank you for the suggestion. The authors have updated the manuscript to include the suggested wording for clarity.